# Nuclear hormone receptor NHR-49 acts in parallel with HIF-1 to promote hypoxia adaptation in *Caenorhabditis elegans*

**Kelsie RS Doering[1,2,3], Xuanjin Cheng[2,3,4], Luke Milburn[5], Ramesh Ratnappan[6], Arjumand Ghazi[6,7], Dana L Miller[5], Stefan Taubert[1,2,3,4]\***

[1]Graduate Program in Medical Genetics, University of British Columbia, Vancouver, Canada; [2]British Columbia Children's Hospital Research Institute, Vancouver, Canada; [3]Centre for Molecular Medicine and Therapeutics, The University of British Columbia, Vancouver, Canada; [4]Department of Medical Genetics, University of British Columbia, Vancouver, Canada; [5]Department of Biochemistry, University of Washington School of Medicine, Seattle, United States; [6]Department of Pediatrics, University of Pittsburgh School of Medicine, Pittsburgh, United States; [7]Departments of Developmental Biology and Cell Biology and Physiology, University of Pittsburgh School of Medicine, Pittsburgh, United States

**Abstract** The response to insufficient oxygen (hypoxia) is orchestrated by the conserved hypoxia-inducible factor (HIF). However, HIF-independent hypoxia response pathways exist that act in parallel with HIF to mediate the physiological hypoxia response. Here, we describe a hypoxia response pathway controlled by *Caenorhabditis elegans* nuclear hormone receptor NHR-49, an orthologue of mammalian peroxisome proliferator-activated receptor alpha (PPARα). We show that *nhr-49* is required for animal survival in hypoxia and is synthetic lethal with *hif-1* in this context, demonstrating that these factors act in parallel. RNA-seq analysis shows that in hypoxia *nhr-49* regulates a set of genes that are *hif-1*-independent, including autophagy genes that promote hypoxia survival. We further show that nuclear hormone receptor *nhr-67* is a negative regulator and homeodomain-interacting protein kinase *hpk-1* is a positive regulator of the NHR-49 pathway. Together, our experiments define a new, essential hypoxia response pathway that acts in parallel with the well-known HIF-mediated hypoxia response.

**\*For correspondence:**
taubert@cmmt.ubc.ca

## Editor's evaluation

The highly conserved protein hypoxia-inducible factor (HIF) is a well-known regulator of animal responses to low-oxygen environments. Using the sophisticated genetic tools of the nematode *C. elegans*, this paper identifies a parallel mechanism, governed by a different conserved transcription factor, that also provides protection from hypoxia. These findings provide important new insight into the complex genetic architecture of the mechanisms that maintain organismal homeostasis in the face of environmental stress.

## Introduction

Organisms are continuously exposed to endogenous and exogenous stresses, from suboptimal temperatures to foreign substances. Thus, an organism's ability to mount specific stress responses, including protecting healthy cells from harm or inducing apoptosis when damage to a cell cannot be overcome, is critical for survival. Hypoxia is a stress that occurs when cellular oxygen levels are too low

for normal physiological functions. It occurs naturally in cells and tissues during development, as well as in many diseases (*Lee et al., 2020*; *Powell-Coffman, 2010*). For example, due to hyperproliferation, inadequate vascularization, and loss of matrix attachment, cancer cells grow in hostile microenvironments featuring hypoxia. Certain cancers thus hijack the hypoxia response to allow growth and metastasis in these harsh conditions (*Rankin and Giaccia, 2016*; *Schito and Semenza, 2016*; *Zhang et al., 2019*), and tumour hypoxia correlates with poor clinical outcome (*Keith and Simon, 2007*). Most prominently, mutations in the tumour suppressor von Hippel–Lindau (VHL), which inhibits the transcription factor hypoxia-inducible factor (HIF), occur in kidney cancers, and the resulting accumulation of HIF drives tumour growth (*Kaelin Jr, 2008*; *Li and Kim, 2011*). In line with a pivotal role of HIF in these cancers are studies showing promising effects of HIF inhibitors in preclinical (*Albadari et al., 2019*; *Chen et al., 2016*; *Cho et al., 2016*) and clinical studies (*Fallah and Rini, 2019*). However, a better understanding of the transcriptional hypoxia adaptation pathway is needed to pinpoint new drug targets and gain a deeper insight into how cells, tissues, and organisms cope with hypoxia.

The pathways that regulate the response to hypoxia are evolutionarily conserved from the nematode worm *Caenorhabditis elegans* to humans. As in mammals, a key pathway in *C. elegans* involves the transcription factor HIF-1, which is critical for the cellular responses to and the defence against hypoxia (*Choudhry and Harris, 2018*; *Jiang et al., 2001*). To survive hypoxia, animals activate the EGL-Nine homolog (EGLN)–VHL-HIF pathway (*egl-9–vhl-1–hif-1* in *C. elegans*). In normoxic conditions (21% $O_2$), HIF-1 is degraded and thus inactive. This occurs when EGL-9 adds a hydroxyl group onto a proline residue within HIF-1. The hydroxylated proline promotes binding of the E3 ubiquitin ligase VHL-1, leading to poly-ubiquitination and proteasomal degradation of HIF-1. However, in hypoxic conditions, EGL-9 is rendered inactive; hence, HIF-1 is stabilized and activates a hypoxia adaptation gene expression program (*Epstein et al., 2001*; *Powell-Coffman, 2010*).

Although the responses controlled by the HIF-1 master regulator are most studied, evidence for parallel transcriptional programs in hypoxia exists, from *C. elegans* to mammalian organisms. For example, the transcription factor B lymphocyte-induced maturation protein 1 (BLMP-1) has a *hif-1*-independent hypoxia regulatory role in *C. elegans* (*Padmanabha et al., 2015*), as does the conserved nuclear hormone receptor (NHR) oestrogen-related receptor (dERR) in *Drosophila melanogaster* (*Li et al., 2013*), and the cargo receptor sequestosome 1 (SQSTM1/p62) in mammals (*Pursiheimo et al., 2009*). Thus, despite the evolutionarily conserved and important role of the HIF family, robust and effective hypoxia adaptation requires an intricate network of transcription factors that act in concert. Compared to HIF, there is far less known about the mechanisms by which these pathways contribute to the hypoxia response.

*C. elegans* NHR-49 is a transcription factor orthologous to mammalian hepatocyte nuclear factor 4 (HNF4) and peroxisome proliferator-activated receptor α (PPARα) (*Lee et al., 2016*). Similar to these NHRs, it controls lipid metabolism by activating genes involved in fatty acid desaturation and mitochondrial β-oxidation (*Pathare et al., 2012*; *Van Gilst et al., 2005a*). By maintaining lipid homeostasis, NHR-49 is able to extend lifespan, a phenotype often associated with stress resistance (*Burkewitz et al., 2015*; *Ratnappan et al., 2014*). In addition to regulating lipid metabolism, NHR-49 also regulates putative xenobiotic detoxification genes in a dietary restriction-like state and during starvation (*Chamoli et al., 2014*; *Goh et al., 2018*), is required for resistance to oxidative stress (*Goh et al., 2018*), and activates innate immune response programs upon infection of *C. elegans* with *Staphylococcus aureus* (*Wani et al., 2021*), *Pseudomonas aeruginosa* (*Naim et al., 2021*), and *Enterococcus faecalis* (*Dasgupta et al., 2020*). Moreover, a recent report showed that *nhr-49* is required to increase expression of the catechol-O-methyl-transferase *comt-5* in hypoxia, acting downstream of the hypoxia-inhibited receptor tyrosine kinase *hir-1* (*Vozdek et al., 2018*). However, the role of *nhr-49* in hypoxia and how it intersects with *hif-1* have not been explored.

The detoxification gene flavin mono-oxygenase 2 (*fmo-2*) is induced in many of the aforementioned stresses in an *nhr-49*-dependent manner (*Dasgupta et al., 2020*; *Goh et al., 2018*; *Wani et al., 2021*). Interestingly, *fmo-2* is also a *hif-1*-dependent hypoxia response gene (*Leiser et al., 2015*; *Shen et al., 2005*), but its dependence on *nhr-49* in hypoxia is not known. We hypothesized that *nhr-49* may play a role in the worm hypoxia response, in part by regulating *fmo-2* expression. Here, we show that *nhr-49* is not only required to induce *fmo-2*, but controls a broad transcriptional response to hypoxia, including the induction of autophagy, a process required within the *nhr-49* pathway for survival in hypoxia. Our epistasis experiments indicate that *nhr-49* is functionally required independently of *hif-1*

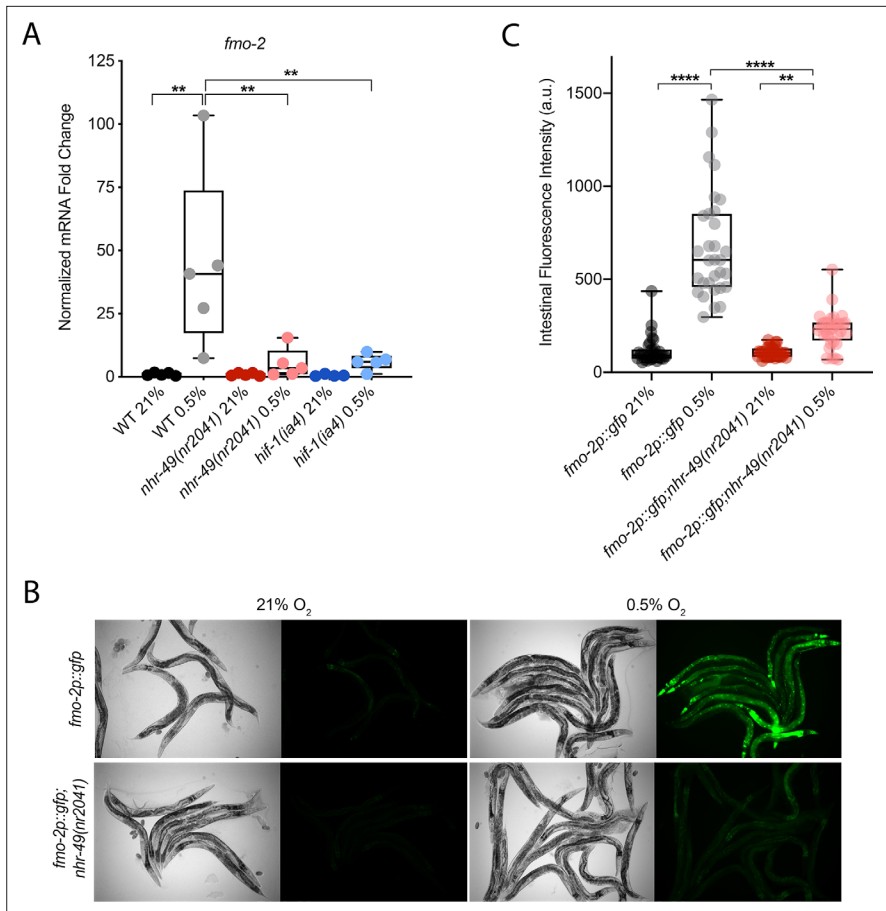

**Figure 1.** *nhr-49* regulates *fmo-2* induction following exposure to hypoxia. (**A**) The graph indicates fold changes of mRNA levels (relative to unexposed wild-type) in L4 wild-type, *nhr-49(nr2041)*, and *hif-1(ia4)* animals exposed to room air (21% $O_2$) or 0.5% $O_2$ for 3 hr (n = 5). **p<0.01 (two-way ANOVA corrected for multiple comparisons using the Tukey method). (**B**) Representative micrographs show *fmo-2p::gfp* and *fmo-2p::gfp;nhr-49(nr2041)* adult animals in room air or following 4 hr exposure to 0.5% $O_2$ and 1 hr recovery in 21% $O_2$. (**C**) The graph shows the quantification of intestinal GFP levels in *fmo-2p::gfp* and *fmo-2p::gfp;nhr-49(nr2041)* animals following 4 hr exposure to 0.5% $O_2$ and 1 hr recovery in 21% $O_2$ (three repeats totalling >30 individual animals per genotype). **p<0.01, ****p<0.0001 (two-way ANOVA corrected for multiple comparisons using the Tukey method). WT: wild-type. See ***Source data 1*** for (**A**) and (**C**).

in hypoxia. Finally, we identify the protein kinase homeodomain-interacting protein kinase 1 (*hpk-1*) as an upstream activator and the transcription factor *nhr-67* as a repressor of the *nhr-49* hypoxia response pathway. Together, our data define NHR-49 as a core player in a novel hypoxia response pathway that acts in parallel with HIF-1.

## Results

### NHR-49 is required to induce the expression of *fmo-2* in hypoxia

*C. elegans fmo-2* is induced by oxidative stress, starvation, and pathogen infection in an *nhr-49-*dependent fashion (***Dasgupta et al., 2020***; ***Goh et al., 2018***; ***Wani et al., 2021***). *fmo-2* expression is also induced in a *hif-1*-dependent manner during hypoxia (0.1% $O_2$; ***Leiser et al., 2015***; ***Shen et al., 2005***). To test whether *nhr-49* regulates *fmo-2* expression in hypoxia, we quantified *fmo-2* mRNA levels in normoxia (21% $O_2$) and hypoxia (0.5% $O_2$) by quantitative reverse transcription PCR (qRT-PCR) in wild-type and mutant animals. The *nr2041* allele deletes portions of both the DNA-binding domain and the ligand-binding domain of *nhr-49* and is a predicted molecular null allele (***Van Gilst et al., 2005b***). The *ia4* allele deletes exons 2–4 of *hif-1* and is also a predicted null allele (***Jiang et al., 2001***).

In wild-type animals, *fmo-2* transcript levels increased approximately 40-fold in hypoxia, but this induction was blocked in both *nhr-49(nr2041)* and *hif-1(ia4)* mutant animals (*Figure 1A*). Experiments using a transgenic strain expressing a transcriptional *fmo-2p::gfp* reporter (*Goh et al., 2018*) corroborated these observations in vivo. In normoxia, this reporter is weakly expressed in some neurons and in the intestine of transgenic animals, but expression was significantly elevated in the intestine of transgenic animals in hypoxia (*Figure 1B and C*). High pharyngeal expression made it difficult to quantify neuronal *fmo-2p::gfp* in hypoxia. Consistent with our qRT-PCR data, loss of *nhr-49* abrogated the increase in intestinal upregulation of *fmo-2p::gfp* animals following hypoxia exposure. We conclude that *nhr-49* is required to induce *fmo-2* in hypoxia.

## *nhr-49* is required throughout the *C. elegans* life cycle to promote hypoxia resistance in parallel with *hif-1*

Wild-type *C. elegans* embryos can survive a 24 hr exposure to environments with as little as 0.5% $O_2$, dependent on the presence of *hif-1* (*Jiang et al., 2001*; *Nystul and Roth, 2004*). We wanted to determine if *nhr-49*, like *hif-1*, is functionally required for animal survival during hypoxia. We first assessed the ability of embryos to survive for 24 hr in 0.5% $O_2$ and then recover to the L4 or later stage when placed back in normoxia for 65 hr. We found that 86% of wild-type embryos reached at least the L4 stage, while only 25% of *nhr-49* and *hif-1* null mutant animals reached at least the L4 stage by that time (*Figure 2A*). The sensitivity of *nhr-49* null mutant animals to hypoxia is specific to the loss of *nhr-49*, as transgenic re-expression of NHR-49 from its endogenous promoter rescues this phenotype (see below). This shows that, like *hif-1*, *nhr-49* is required for embryo survival in hypoxia.

Next, we asked whether *nhr-49* acts in the *hif-1* hypoxia response pathway or in a separate, parallel response pathway. To address this question, we generated an *nhr-49(nr2041);hif-1(ia4)* double null mutant. We observed that less than 2% of *nhr-49;hif-1* double null mutants reached at least the L4 stage following hypoxia exposure (*Figure 2A*). This suggests that *nhr-49* and *hif-1* act in separate, genetically parallel hypoxia response pathways.

To determine if *nhr-49* and *hif-1* are required for larval development in hypoxia, we exposed newly hatched, first stage (L1) larvae to hypoxia for 48 hr. Following this treatment, 95% of wild-type animals reached at least the L4 stage (*Figure 2B*). In contrast, only 19% of *nhr-49* and only 20% of *hif-1* mutant animals, respectively, reached at least the L4 stage, and no *nhr-49;hif-1* double null mutant animals survived and developed to L4 (*Figure 2B*). Together, these results show that *nhr-49* is required for worm adaptation to hypoxia in a pathway parallel to that of *hif-1* both during embryogenesis and post-embryonically.

In normal conditions, *nhr-49* null animals have a shortened lifespan (*Van Gilst et al., 2005b*). This raised the concern that the defects observed in hypoxia may be an indirect consequence of NHR-49's normal developmental roles. To test whether the effects observed above were due to a specific requirement for *nhr-49* in the hypoxia response, we studied worm development in normoxia. We found that loss of *nhr-49* did not affect animal survival from the embryo to at least the L4 stage at 21% $O_2$ (*Figure 2—figure supplement 1A*, *Supplementary file 1*). Additionally, although *nhr-49* null mutants develop slower than wild-type animals at 21% $O_2$, the majority of animals (88%) develop to at least the L4 stage after 48 hr, which is a significantly higher portion than develop to at least the L4 stage in 0.5% $O_2$ (19%; *Figure 2—figure supplement 1B*, *Supplementary file 2*). Together, these data show that although *nhr-49* null mutants display mild developmental defects in normoxia, the phenotypes observed are due to the requirement for *nhr-49* specifically during hypoxia.

## *nhr-49* is dispensable for survival in hydrogen sulfide

To assess whether *nhr-49* is involved in other responses requiring *hif-1*, we next asked if it was required for adaptation to hydrogen sulfide ($H_2S$). $H_2S$ is produced endogenously and is an important signalling molecule in animals, including in *C. elegans* (*Li et al., 2011*). However, exposure to high levels of hydrogen sulfide can be lethal. As in the hypoxia response, *hif-1* is a master regulator of the transcriptional response to exogenous hydrogen sulfide, and *hif-1* is required for worm survival in 50 ppm $H_2S$ (*Budde and Roth, 2010*; *Miller et al., 2011*). In contrast, we found that *nhr-49* null mutants survive exposure to 50 ppm $H_2S$ as well as wild-type control animals (*Figure 2C*). This suggests that the requirement for *nhr-49* is stress specific, and that *nhr-49* does not participate in all *hif-1*-dependent stress responses. This is consistent with previous observations that the *hif-1*-dependent changes in

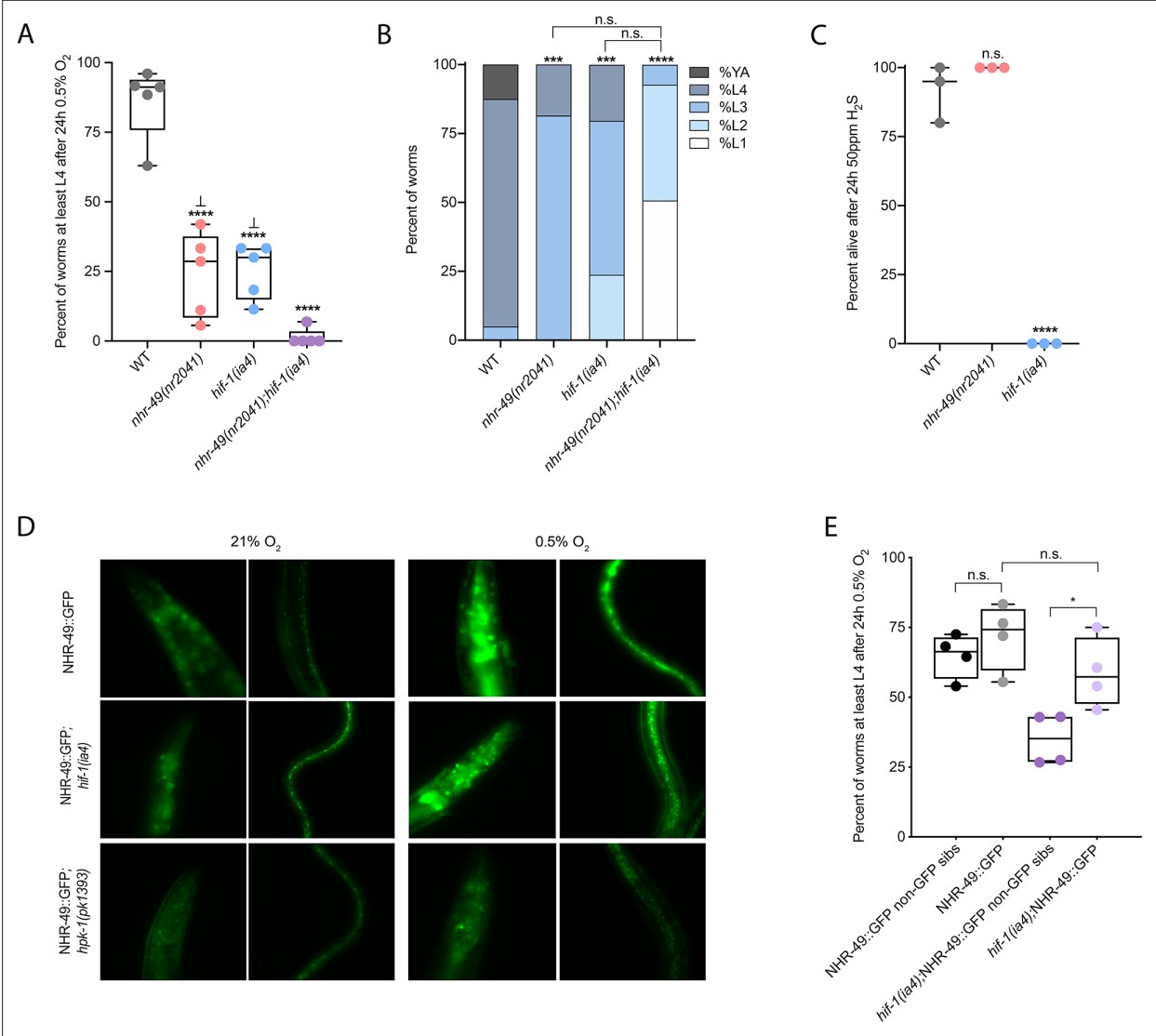

**Figure 2.** *nhr-49* and *hif-1* act in parallel hypoxia response pathways at two stages of the worm life cycle. (**A**) The graph shows the average population survival of wild-type, *nhr-49(nr2041)*, *hif-1(ia4)*, and *nhr-49(nr2041);hif-1(ia4)* worm embryos exposed for 24 hr to 0.5% $O_2$ and then allowed to recover at 21% $O_2$ for 65 hr, counted as the ability to reach at least the L4 stage (five repeats totalling >100 individual animals per genotype). ****p<0.0001 vs. wild-type animals, ⊥p<0.05 vs. *nhr-49(nr2041);hif-1(ia4)* (ordinary one-way ANOVA corrected for multiple comparisons using the Tukey method). (**B**) The graph shows the average developmental success of wild-type, *nhr-49(nr2041)*, *hif-1(ia4)*, and *nhr-49(nr2041);hif-1(ia4)* larval worms following 48 hr exposure to 0.5% $O_2$ from L1 stage (four repeats totalling >60 individual animals per genotype). ***p<0.001, ****p<0.0001 percent L4 or older vs. wild-type animals (ordinary one-way ANOVA corrected for multiple comparisons using the Tukey method). (**C**) The graph shows the average population survival of wild-type, *nhr-49(nr2041)*, and *hif-1(ia4)* L4 animals following 24 hr exposure to 50 ppm hydrogen sulfide (three repeats totalling 60 individual animals per strain). ****p<0.0001 vs. wild-type animals (ordinary one-way ANOVA corrected for multiple comparisons using the Tukey method). (**D**) High-magnification images show *nhr-49p::nhr-49::gfp* adult worms in wild-type, *hif-1(ia4)*, and *hpk-1(pk1393)* backgrounds exposed to room air or following 4 hr exposure to 0.5% $O_2$ and 1 hr recovery in 21% $O_2$. Expression is seen in the head, intestine, and hypodermal seam cells (additional repeats in *Figure 2—figure supplement 1E and F*) (**E**) The graph shows the average population survival of *nhr-49p::nhr49::gfp* and *nhr-49p::nhr49::gfp;hif-1(ia4)* animals and their respective non-GFP sibling embryos exposed for 24 hr to 0.5% $O_2$ and then allowed to recover at 21% $O_2$ for 65 hr, counted as the ability to reach at least the L4 stage (four repeats totalling >100 individual animals per genotype). *p<0.05 (ordinary one-way ANOVA corrected for multiple comparisons using the Tukey method). YA : young adult; n.s.: not significant; WT: wild-type. See *Source data 1* for (**A–C, E**).

The online version of this article includes the following figure supplement(s) for figure 2:

**Figure supplement 1.** *nhr-49* and *hif-1* mutants do not display major developmental defects in normoxia, and NHR-49::GFP is induced by hypoxia.

gene expression in H$_2$S are quite different than those seen in hypoxia (*Miller et al., 2011*). Additionally, the ability of *nhr-49* mutants to readily adapt to H$_2$S provides further evidence that the mild developmental defects of *nhr-49* null mutants do not render the animal sensitive to all stresses. Instead, our data indicate that *nhr-49*'s requirement for hypoxia survival is due to a specific function for this regulator in this particular stress condition.

## NHR-49 overexpression compensates for the loss of *hif-1* in hypoxia survival

Next, we asked whether the *hif-1* and *nhr-49* pathways crosstalk in hypoxia. First, we studied NHR-49 levels in hypoxia using the *nhr-49p::nhr-49::gfp* translational reporter, which expresses a GFP-tagged, full-length NHR-49 fusion protein from its own promoter from an extra-chromosomal array (henceforth referred to as NHR-49::GFP; *Ratnappan et al., 2014*). Interestingly, we observed an induction of NHR-49::GFP signal in animals exposed to hypoxia (*Figure 2D*; see also below). Next, we crossed the NHR-49::GFP transgene into the *hif-1* mutant background; in the resulting strain, the NHR-49::GFP induction resembled that seen in the wild-type background (*Figure 2D*, *Figure 2—figure supplement 1C and D*). Higher-magnification images showed that the NHR-49 induction in hypoxia was similar in the head, intestine, and hypodermal seam cells in the *hif-1* null background (*Figure 2D*, *Figure 2—figure supplement 1E and F*). In sum, loss of *hif-1* does not appear to induce NHR-49 protein levels.

To further explore *nhr-49* and *hif-1* crosstalk, we tested if NHR-49 was able to rescue the hypoxia survival defects of the *hif-1* null mutant. Although NHR-49 overexpression did not protect wild-type embryos from hypoxia (wild-type vs. non-GFP siblings), NHR-49 overexpression restored *hif-1* null embryo survival to the level seen in wild-type animals (*Figure 2E*). Thus, overexpression of NHR-49 compensates for the loss of *hif-1*, further suggesting that these two transcription factors act in parallel pathways.

## The *nhr-49*-dependent transcriptional response to hypoxia includes *hif-1*-independent genes

To delineate the genes and biological processes regulated by NHR-49 in hypoxia, we analysed whole-animal transcriptomes of wild-type, *nhr-49*, and *hif-1* mutant animals before and after a 3 hr exposure to hypoxia (0.5% O$_2$) using RNA-sequencing (RNA-seq; *Figure 3A and B*, *Figure 3—figure supplement 1A*). Consistent with published microarray data (*Shen et al., 2005*), we found that hypoxia in wild-type animals upregulated more genes (718) than it downregulated (339); collectively, we refer to these as hypoxia-responsive genes (1,057) (*Figure 3A*; false discovery rate [FDR] < 0.05, fold regulation ≥2). Despite different experimental setups (harvest stages, oxygen percentage, gene expression profiling technique), we found a significant overlap in hypoxia-induced genes when comparing our data to the data from Shen et al. (*Figure 3—figure supplement 1B*). Our data also identified several experimentally confirmed hypoxia-inducible genes, such as *egl-9*, *phy-2*, *nhr-57*, F22B5.4, and *fmo-2* (*Bishop et al., 2004*; *Shen et al., 2005*), validating our approach (*Figure 3—figure supplement 1B and C*).

Next, we performed functional enrichment profiling to elucidate the biological pathways and processes governed by hypoxia-responsive genes. In wild-type animals, hypoxia-induced genes function mainly in pathways such as detoxification, response to heavy metal stress, and autophagy, whereas hypoxia-repressed genes play roles in processes such as amino acid transport (*Figure 3—figure supplement 1D*). Interestingly, a set of genes involved in amino acid metabolism was induced while another set was repressed by hypoxia, whereas genes involved in insulin-related metabolism were exclusively repressed (*Figure 3—figure supplement 1E*).

Then, we performed intersection analysis to identify genes that require *nhr-49* and/or *hif-1* to respond to hypoxia (*Figure 3A and B*). We found that 315 upregulated genes (of 718 upregulated in wild type) failed to be upregulated and 177 downregulated genes (of 339 downregulated in wild type) failed to be downregulated in *nhr-49* mutants (*Figure 3A and B*); collectively we call these *nhr-49*-dependent genes. Of these *nhr-49*-dependent genes, 83 of the upregulated and 51 of the downregulated genes were *hif-1*-independent (*Figure 3A and B*). In line with our above data, *fmo-2* was induced in an *nhr-49*-dependent manner (*Figure 3C*). However, although our qRT-PCR data (*Figure 1A*) show that *fmo-2* induction is dependent on *hif-1*, our RNA-seq analysis excluded *fmo-2* from the *hif-1*-dependent set because it retained more than twofold induction in hypoxia vs. normoxia

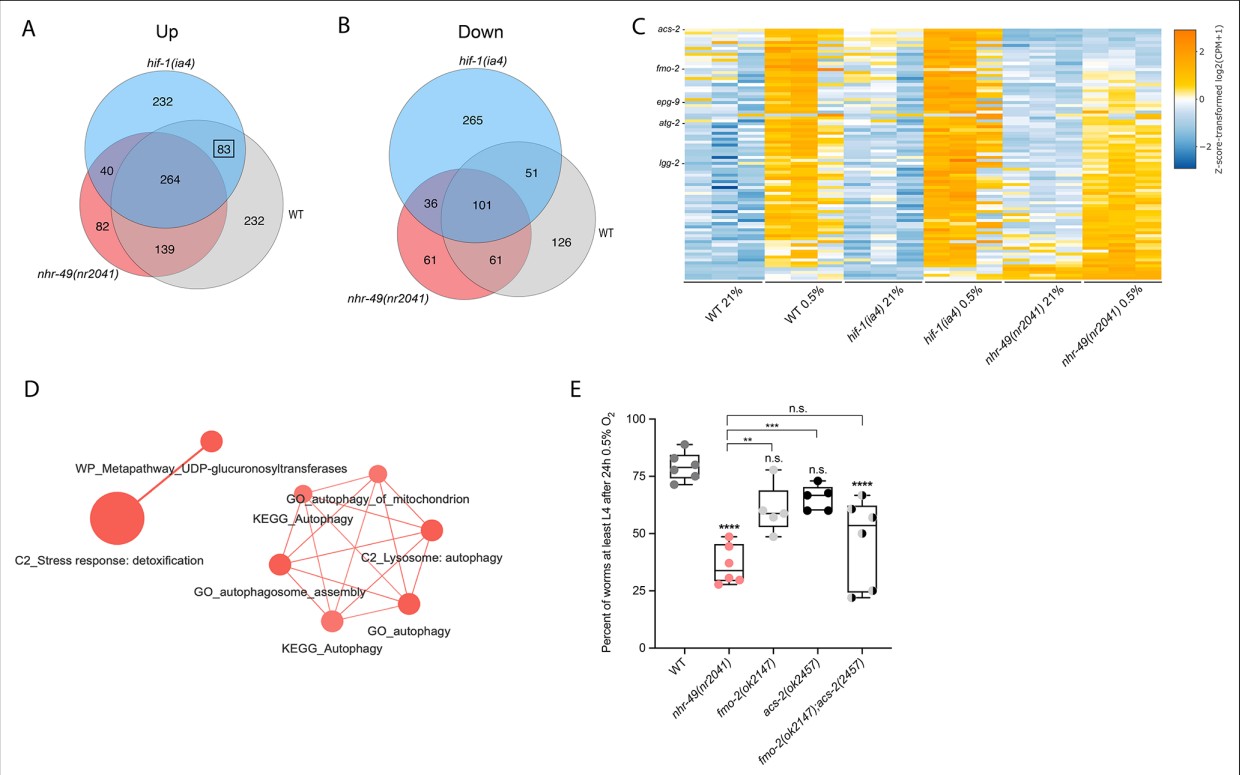

**Figure 3.** RNA-seq reveals an *nhr-49*-dependent transcriptional program in hypoxia. (**A, B**) Venn diagrams show the overlap of genes regulated by hypoxia (3 hr 0.5% $O_2$; vs. normoxia 21% $O_2$) in wild-type, *nhr-49(nr2041)*, and *hif-1(ia4)* animals. Numbers indicate the number of differentially (FDR < 0.05, |logFC| ≥ 1) expressed genes in wild-type (grey), *nhr-49* (red), and/or *hif-1* (blue) animals in hypoxia, with hypoxia-upregulated genes in (**A**) and hypoxia-downregulated genes in (**B**). Genes upregulated by hypoxia in wild-type include 83 + 264 + 139 + 232 = 718 total genes. Of these 718 genes, 315 genes are not induced in *nhr-49* animals, i.e., they require *nhr-49* for induction; these are composed of 83 genes induced in both wild-type and *hif-1* animals, but not in *nhr-49* animals (grey-blue overlap; these depend only on *nhr-49* but not on *hif-1*; highlighted by black box), and 232 genes induced only in wild-type animals but not in *hif-1* or *nhr-49* animals (grey; i.e., these are co-dependent on *nhr-49* and *hif-1*). Genes downregulated by hypoxia include 177 genes that require *nhr-49* for repression, composed of 51 genes downregulated in wild-type and *hif-1* animals, but not in *nhr-49* animals (grey-blue overlap; these depend on *nhr-49* only); and 126 genes downregulated only in wild-type animals but not in *hif-1* or *nhr-49* animals (grey; i.e., these are co-dependent on *nhr-49* and *hif-1*). (**C**) Heatmap of the expression levels of the 83 genes, which are significantly induced over twofold in 21% $O_2$ vs. 0.5% $O_2$ in wild-type and *hif-1(ia4)* animals, but not in *nhr-49(nr2041)*, i.e., *nhr-49*-dependent hypoxia response genes. Genes along the y-axis are coloured in each repeat based on their z-scores of the log2-transformed counts per million (CPM) plus 1. Notable genes are highlighted. (**D**) Network view of the enriched functional categories among the 83 genes, which are significantly induced over twofold in 21% $O_2$ vs. 0.5% $O_2$ in wild-type and *hif-1(ia4)* animals, but not in *nhr-49(nr2041)*. Edges represent significant gene overlap as defined by a Jaccard coefficient larger than or equal to 25%. The dot size reflects the number of genes in each functional category; colour intensity reflects statistical significance (−log10 p-value). (**E**) The graph shows the average population survival of wild-type, *nhr-49(nr2041)*, *fmo-2(ok2147)*, *acs-2(ok2457)*, and *fmo-2(ok2147);acs-2(ok2457)* embryos following 24 hr exposure to 0.5% $O_2$, then allowed to recover at 21% $O_2$ for 65 hr, and counted as the ability to reach at least L4 stage (five or more repeats totalling >100 individual animals per strain). **p<0.01, ****p<0.0001 vs. wild-type animals. Comparison of single mutants to *fmo-2(ok2147);acs-2(ok2457)* not significant (ordinary one-way ANOVA corrected for multiple comparisons using the Tukey method). n.s.: not significant; WT: wild-type; FDR: false discovery rate; |logFC|: log2-transformed fold change. See ***Source data 1*** for (**E**).

The online version of this article includes the following figure supplement(s) for figure 3:

**Figure supplement 1.** RNA-seq reveals several discrete hypoxia-responsive transcriptional programs.

**Figure supplement 2.** *nhr-49* regulates *acs-2* induction following exposure to hypoxia.

---

(***Figure 3—figure supplement 1C***). This suggests that although *fmo-2* induction is somewhat dependent on *hif-1*, it requires *nhr-49*. Thus, although many hypoxia-responsive genes are controlled by both transcription factors, a subset is *nhr-49*-dependent but *hif-1*-independent.

Next, we functionally profiled the 83 genes that exclusively require *nhr-49* but not *hif-1* for induction in hypoxia using functional enrichment analysis (***Figure 3C***, ***Supplementary file 3a***). We found that autophagy and detoxification genes were significantly enriched (***Figure 3D***), suggesting a requirement for *nhr-49* to regulate these processes in hypoxia. Interestingly, a separate set of detoxification

genes was dependent only on *hif-1* (*Figure 3—figure supplement 1F and G*, *Supplementary file 3b*), and a third set of detoxification genes was independent of both *nhr-49* and *hif-1* (*Figure 3—figure supplement 1H and I*, *Supplementary file 3c*). This suggests that there may be an additional transcription factor(s) regulating this process in hypoxia.

Our RNA-seq data revealed that the acyl-CoA synthetase gene *acs-2* is induced in response to hypoxia in an *nhr-49*-dependent manner (*Figure 3C*, *Supplementary file 3a*). ACS-2 acts in the first step of mitochondrial fatty acid β-oxidation and is strongly induced by NHR-49 during starvation and following exposure to *E. faecalis* (*Dasgupta et al., 2020*; *Van Gilst et al., 2005a*). To validate our RNA-seq data, we quantified *acs-2* expression via qRT-PCR. Following hypoxia exposure, *acs-2* transcript levels increased approximately 12-fold, and this induction was blocked in the *nhr-49* null mutant, but not the *hif-1* null mutant (*Figure 3—figure supplement 2A*). We used a transgenic strain expressing a transcriptional *acs-2p::gfp* reporter to study this regulation in vivo (*Burkewitz et al., 2015*). This reporter showed moderate GFP expression in the body of animals under normoxia, but expression increased substantially in the intestine following exposure to hypoxia (*Figure 3—figure supplement 2B and C*). Consistent with our RNA-seq and qRT-PCR data, loss of *nhr-49* blocked transcriptional activation via the *acs-2* promoter as GFP was weaker in the intestines of these animals following hypoxia exposure (*Figure 3—figure supplement 2B and C*). Collectively, these data provide in vivo evidence that *nhr-49* is specifically required, and that *hif-1* is dispensable, for induction of *acs-2* in hypoxia.

## Autophagy genes are critical downstream targets of *nhr-49* in hypoxia

Next, we wanted to determine which of *nhr-49*'s downstream transcriptional targets are functionally important for animal survival in hypoxia. We first assessed the ability of *fmo-2(ok2147)* and *acs-2(ok2457)* embryos to survive hypoxia as both genes are strongly induced by hypoxia in an *nhr-49*-dependent manner. Individually, loss of either *fmo-2* (60% of embryos develop to at least the L4 stage) or *acs-2* (65%) did not significantly decrease embryo viability compared to wild type (79%) (*Figure 3E*). However, simultaneous loss of both *fmo-2* and *acs-2* resulted in a significant decrease in survival after hypoxia (47%). None of the mutant animals showed embryo viability defects in normoxia, indicating that the phenotypes observed were specifically due to the requirement of these genes in hypoxia survival (*Figure 4—figure supplement 1A*, *Supplementary file 1*). These data suggest that *fmo-2* and *acs-2* each contribute only modestly to worm survival in hypoxia and are likely not the main factors contributing to *nhr-49*'s importance in survival to this stress. This resembles previous observations that mutations that disrupt individual *hif-1*-responsive genes show only minor defects in hypoxia survival (*Shen et al., 2005*).

Our RNA-seq analysis revealed autophagy as a major biological process modulated by *nhr-49* (*Figure 3D*). Notably, *C. elegans* show sensitivity to anoxia when the autophagy pathway is disrupted (*Samokhvalov et al., 2008*), and autophagy is upregulated in anoxia (*Chapin et al., 2015*). However, the responses to anoxia and hypoxia are mediated by different regulatory pathways (*Nystul and Roth, 2004*), and it thus was not a priori clear whether autophagy is also required for hypoxia resistance. First, to validate our RNA-seq results, we examined the expression of three autophagy genes with transcriptional (promoter::gfp) reporters. Hypoxia significantly induced GFP fluorescence in worms bearing *lgg-1p::gfp*, *atg-2p::gfp*, or *epg-3p::gfp* reporters (*Figure 4A and B*, *Figure 4—figure supplement 1B and C*). Consistent with our RNA-seq results, *nhr-49* was required for these inductions, whereas *hif-1* was not.

To test whether *nhr-49* is required for autophagosome formation in hypoxia, we studied the widely used LGG-1::GFP reporter (*Das et al., 2017*; *Palmisano and Meléndez, 2016*; *Samokhvalov et al., 2008*; *Zhang et al., 2015*), wherein GFP is tagged to the C-terminus of the autophagosome assembly factor LGG-1 (Atg8 in mammals, LC3 in yeast). In wild-type animals, a 5 hr exposure to hypoxia significantly increased the number of LGG-1::GFP foci in seam cells compared to normoxia exposure (*Figure 4C and D*). Critically, loss of *nhr-49* abrogated the increase in LGG-1::GFP foci following hypoxia exposure, whereas loss of *hif-1* did not. This shows that autophagosome formation in hypoxia is dependent on *nhr-49*, but independent of *hif-1*.

To determine if upregulation of autophagy by *nhr-49* is required for worm survival in hypoxia, we assessed the ability of *lgg-2(tm5755)* and *epg-6(tm8366)* mutant embryos to survive hypoxia. Similar to *nhr-49* mutant animals, only 41% of *lgg-2(tm5755)* and 44% of *epg-6(tm8366)* mutant embryos developed to L4 following exposure to hypoxia (*Figure 4E and F*). Next, we used epistasis analysis

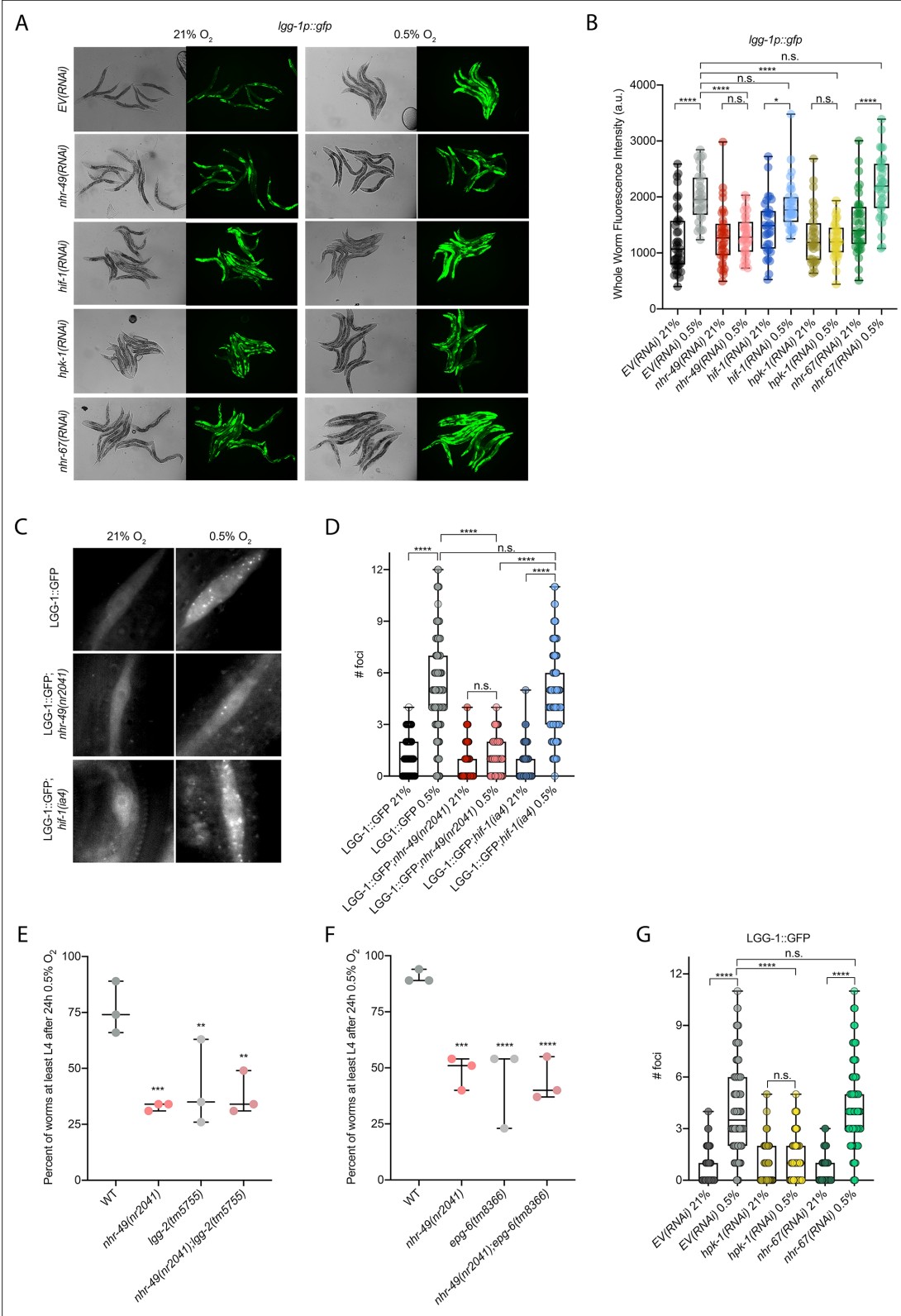

**Figure 4.** *nhr-49* is required to induce autophagy in response to hypoxia. (**A, B**) The figure shows representative micrographs (**A**) and whole-worm GFP quantification (**B**) of *lgg-1p::gfp* adult animals fed EV, *nhr-49*, *hif-1*, *hpk-1*, or *nhr-67* RNAi in room air or following 4 hr exposure to 0.5% $O_2$ and 1 hr recovery in 21% $O_2$ (three repeats totalling >30 individual animals per genotype). *$p<0.05$, ****$p<0.0001$ (two-way ANOVA corrected for multiple comparisons using the Tukey method). (**C, D**) The figure shows representative micrographs (**C**) and quantification (**D**) of LGG-1::GFP foci in individual

*Figure 4 continued on next page*

*Figure 4 continued*

hypodermal seam cells in L3 animals in the wild-type, *nhr-49(nr2041)*, and *hif-1(ia4)* backgrounds, kept in room air or exposed to 5 hr 0.5% O₂ (three repeats totalling >110 individual seam cells in at least 15 individual animals per genotype). Micrograph brightness and contrast are matched within genotype, and unmatched between genotypes. ****p<0.0001 (two-way ANOVA corrected for multiple comparisons using the Tukey method). (**E, F**) The graphs show average population survival of wild-type, *nhr-49(nr2041)*, (**E**) *lgg-2(tm5755)* and *nhr-49(nr2041);lgg-2(tm5755)*, and (**F**) *epg-6(tm8366)* and *nhr-49(nr2041);epg-6(tm8366)* animal embryos exposed for 24 hr to 0.5% O₂ and then allowed to recover at 21% O₂ for 65 hr, counted as the ability to reach at least the L4 stage (three repeats totalling >100 individual animals per genotype). **p<0.01, ***p<0.001, **** p<0.0001 vs. wild-type animals (ordinary one-way ANOVA corrected for multiple comparisons using the Tukey method). (**G**) Quantification of LGG-1::GFP foci in individual hypodermal seam cells in L3 animals in second-generation wild-type animals fed EV, *hpk-1*, or *nhr-67* RNAi, kept in room air or exposed to 5 hr 0.5% O₂ (three repeats totalling >110 individual seam cells in at least 15 individual animals per genotype). ****p<0.0001 (two-way ANOVA corrected for multiple comparisons using the Tukey method). n.s.: not significant; WT: wild-type. See *Source data 1* for (**B, D, E– G**).

The online version of this article includes the following figure supplement(s) for figure 4:

**Figure supplement 1.** Mutants of downstream transcriptional targets of *nhr-49* in hypoxia do not display functional defects in normoxia, and autophagy genes are regulated by and act in the *nhr-49* hypoxia response pathway.

to test whether genes involved in autophagy act in the *nhr-49* pathway to promote worm survival in hypoxia. We observed that *nhr-49;lgg-1* (38%) and *nhr-49;epg-6* (44%) double mutants showed similar survival as does each single null mutant, suggesting that these autophagy genes act in the same pathway as *nhr-49* (***Figure 4E and F***). Each mutant showed normal development from embryo to L4 in normoxia (***Figure 4—figure supplement 1D and E***, ***Supplementary file 1***), indicating that the phenotypes observed were specifically due to the requirement of these genes in hypoxia survival. To corroborate these results, we also depleted several autophagy genes using feeding RNA interference (RNAi) in the wild-type and *nhr-49* null mutant backgrounds and assessed the ability of these embryos to survive hypoxia. RNAi-mediated knockdown of the autophagy genes *atg-10* (28%), *atg-7* (41%), *bec-1* (27%), and *epg-3* (38%) caused significant sensitivity to hypoxia in the wild-type background compared to the empty vector (EV) control RNAi treatment (79%; ***Figure 4—figure supplement 1F***). Importantly, the sensitivity of animals did not significantly change when these genes were knocked down in the *nhr-49* null mutant background (32, 25, 13, and 13%, respectively, vs. *nhr-49(null);EV(RNAi)*

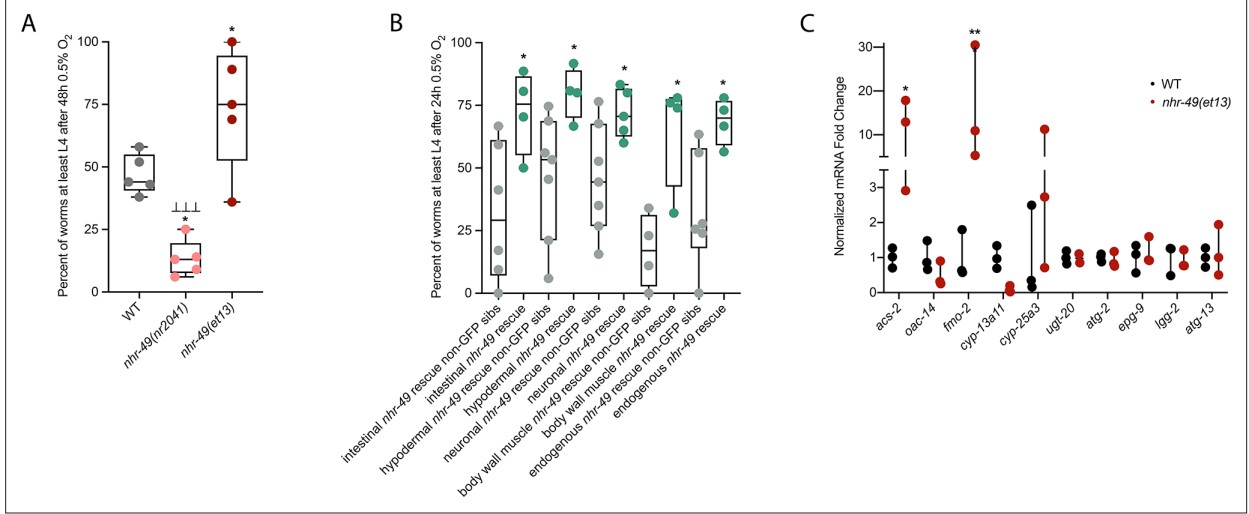

**Figure 5.** *nhr-49* is sufficient to promote survival in hypoxia and induce some hypoxia response genes. (**A**) The graph shows the average population survival of wild-type, *nhr-49(nr2041)*, and *nhr-49(et13)* worm embryos following 48 hr exposure to 0.5% O₂, then allowed to recover at 21% O₂ for 42 hr, and counted as the ability to reach at least L4 stage (five repeats totalling >100 individual animals per strain). *p<0.05 vs. wild-type animals, ⊥⊥⊥p<0.001 vs. *nhr-49(et13)* animals (ordinary one-way ANOVA corrected for multiple comparisons using the Tukey method). (**B**) The graph shows the average population survival of *nhr-49* tissue-specific rescue worm embryos following 24 hr exposure to 0.5% O₂, then allowed to recover at 21% O₂ for 65 hr, and counted as the ability to reach at least L4 stage. *glp-19p::nhr-49::gfp* for intestine, *col-12p::nhr-49::gfp* for hypodermis, *rgef-1p::nhr-49::gfp* for neurons, *myo-3p::nhr-49::gfp* for body wall muscle, and *nhr-49p::nhr-49::gfp* for endogenous (four or more repeats totalling >50 individual animals per strain). *p<0.05 vs. matching non-GFP siblings. (**C**) The graph shows fold changes of mRNA levels (relative to wild type) in L4 wild-type and *nhr-49(et13)* animals (n = 3). *p<0.05, ***p<0.001 vs. wild-type animals (ordinary one-way ANOVA corrected for multiple comparisons using the Tukey method). WT: wild-type. See *Source data 1* for (**A–C**).

21%), suggesting that these genes act in the same pathway as *nhr-49*. Depletion of these genes by RNAi alone did not cause impaired development from embryo to L4 in normoxia, indicating that the phenotypes observed were specifically due to the requirement of these genes in hypoxia survival (*Figure 4—figure supplement 1G*, *Supplementary file 1*). Together, these data show that autophagy is a functionally important *nhr-49*-regulated process required for worm survival in hypoxia.

## NHR-49 expression in multiple tissues is sufficient to promote hypoxia survival

To test if *nhr-49* activation is sufficient to promote survival of *C. elegans* in hypoxia, we studied the *nhr-49(et13)* gain-of-function strain, which is sufficient to induce *fmo-2* (*Goh et al., 2018*; *Lee et al., 2016*). After 24 hr of exposure to hypoxia, 86% of wild-type eggs develop to at least L4 stage (*Figure 2A*), but after 48 hr of hypoxia exposure, only 44% develop to at least L4 stage (*Figure 5A*). In contrast, 75% of *nhr-49(et13)* gain-of-function eggs develop to at least L4 stage after 48 hr of hypoxia exposure, indicating that NHR-49 activation is sufficient to improve the population survival of worms in hypoxia.

NHR-49 is expressed in multiple tissues, including the intestine, neurons, muscle, and hypodermis (*Ratnappan et al., 2014*). Neuronal NHR-49 is sufficient to extend lifespan in some contexts and regulates genes in distal tissues (*Burkewitz et al., 2015*), but where the protein acts to regulate the response to hypoxia is unknown. As described above, NHR-49::GFP imaging indicated that NHR-49 protein levels are induced in the intestine, neurons, and hypodermis during hypoxia (*Figure 2D*, *Figure 2—figure supplement 1E and F*). Hence, we asked if the expression of NHR-49 in any one of these tissues could rescue the hypoxia survival defects of the *nhr-49* mutant (*Naim et al., 2021*). Comparing the survival of each tissue-specific NHR-49::GFP rescue strain to their respective non-GFP siblings, we found that expressing *nhr-49* in the intestine, neurons, hypodermis, body wall muscle, or from its endogenous promoter was sufficient to restore population survival to wild-type levels (*Figure 5B*). Taken together, these data suggest that NHR-49 can act in multiple somatic tissues, perhaps cell non-autonomously, to regulate the organismal hypoxia response.

To determine if NHR-49 activity alone is sufficient to induce expression of hypoxia response genes, we tested if the *nhr-49(et13)* gain-of-function mutant strain showed upregulation of *nhr-49*-dependent hypoxia response genes identified in our RNA-seq analysis in the absence of stress (*Figure 5C*). In line with previous findings (*Goh et al., 2018*; *Lee et al., 2016*), *nhr-49* was sufficient to induce *fmo-2* and *acs-2* expression on its own. However, other hypoxia-inducible *nhr-49* regulated genes involved in autophagy and detoxification (*Supplementary file 3a*) were not induced in the *nhr-49(et13)* gain-of-function mutant. It is possible that *nhr-49* regulates autophagy indirectly, or that the *et13* mutation, which has combined gain- and loss-of-function properties (*Lee et al., 2016*), cannot induce these tested autophagy genes. It is also possible that, to induce these genes, NHR-49 acts in concert with another hypoxia-responsive transcription factor or requires binding of a hypoxia-associated signalling molecule or post-translational modification by a hypoxia-regulated factor, which is not activated in the *nhr-49(et13)* mutant. Together, this shows that NHR-49 is sufficient to extend the survival of worms in hypoxia in various tissues, but the gain-of-function strain is only able to induce certain response genes without the presence of stress.

## The nuclear hormone receptor NHR-67 negatively regulates the *nhr-49* hypoxia response

Cellular stress response pathways are intricate networks involving a multitude of proteins. Activation or repression of downstream response genes thus often requires signalling via additional factors such as kinases and transcription factors. To identify factors acting in the *nhr-49*-regulated hypoxia response pathway, we focused on proteins that have previously been reported to physically interact with NHR-49 (*Reece-Hoyes et al., 2013*); such proteins might be regulators of NHR-49. One potential NHR-49-binding protein is NHR-67, the sole *C. elegans* orthologue of the *D. melanogaster* tailless and vertebrate NR2E1 proteins (*Gissendanner et al., 2004*). NHR-67 is important in neural and uterine development (*Fernandes and Sternberg, 2007*; *Verghese et al., 2011*), but a role for this NHR in stress responses has not yet been described. Our RNA-seq data showed that *nhr-67* mRNA expression is modestly increased during hypoxia in wild-type animals and much more substantially induced in the *nhr-49* null background (*Figure 6A*), suggesting a possible regulatory interaction between these two

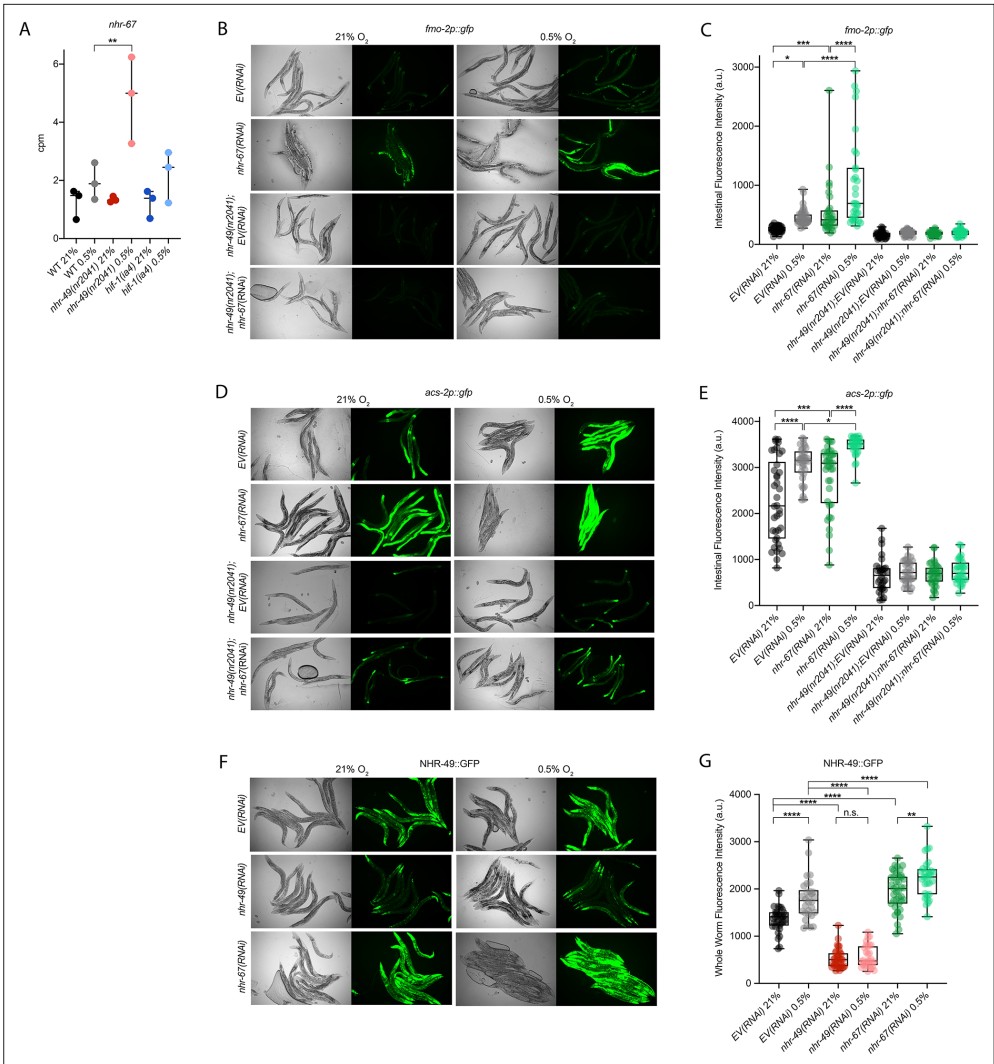

**Figure 6.** *nhr-67* is a negative regulator of the *nhr-49*-dependent hypoxia response pathway. (**A**) The graph shows the average transcript levels in counts per million (CPM) of *nhr-67* mRNA in L4 wild-type, *nhr-49(nr2041)*, and *hif-1(ia4)* animals exposed to 0.5% $O_2$ for 3 hr or kept at 21% $O_2$ (n = 3). **\*\*p <0.01 (two-way ANOVA corrected for multiple comparisons using the Tukey method). (**B–E**) Representative micrographs and quantification of intestinal GFP levels in *fmo-2p::gfp* and *fmo-2p::gfp;nhr-49(nr2041)* (**B, C**) and *acs-2p::gfp* and *acs-2p::gfp;nhr-49(nr2041)* (**D, E**) adult animals fed EV RNAi or *nhr-67* RNAi following 4 hr exposure to 0.5% $O_2$ and 1 hr recovery in 21% $O_2$ (three repeats totalling >30 individual animals per strain). \*p<0.05, \*\*\*p<0.001, \*\*\*\*p<0.0001 (two-way ANOVA corrected for multiple comparisons using the Tukey method). (**F**) Representative micrographs show *nhr-49p::nhr-49::gfp* adult animals fed EV, *nhr-49*, or *nhr-67* RNAi following 4 hr exposure to 0.5% $O_2$ and 1 hr recovery in 21% $O_2$. (**G**) The graph shows quantification of whole-worm GFP levels in *nhr-49p::nhr-49::gfp* animals fed EV, *nhr-49*, or *nhr-67* RNAi following 4 hr exposure to 0.5% $O_2$ and 1 hr recovery in 21% $O_2$ (three or more repeats totalling >30 individual animals per strain). \*\*\*\*p<0.0001 (two-way ANOVA corrected for multiple comparisons using the Tukey method). n.s.: not significant; WT: wild type. See ***Source data 1*** for (**A, C, E, G**).

The online version of this article includes the following figure supplement(s) for figure 6:

**Figure supplement 1.** *nhr-67* is functionally required for survival in hypoxia and acts in the *nhr-49* pathway.

NHRs in hypoxia. To explore this interaction further, we used feeding RNAi to knock down *nhr-67* in normoxia and hypoxia, and observed how this affected the expression of the *fmo-2p::gfp* and *acs-2p::gfp* transcriptional reporters. Compared to *EV(RNAi)*, knockdown of *nhr-67* significantly induced both reporters even in the absence of stress, suggesting a repressive role for *nhr-67* on these genes (***Figure 6B–E***). In hypoxia, *nhr-67(RNAi)* resulted in even higher expression of these reporters. In both normoxia and hypoxia, increased expression of the reporters was dependent on *nhr-49* as loss

of *nhr-49* abrogated the GFP induction (*Figure 6B–E*). The *nhr-49(et13)* gain-of-function mutation is sufficient to induce expression of the *fmo-2p::gfp* reporter in non-stressed conditions (*Goh et al., 2018*), although it does not alter *nhr-67* expression under normoxic conditions (*Figure 6—figure supplement 1A*). Knockdown of *nhr-67* further increased the expression of the *fmo-2p::gfp* reporter in the *nhr-49(et13)* background in both normoxia and hypoxia (*Figure 6—figure supplement 1B and C*). Together, these data suggest that *nhr-67* negatively regulates the expression of the hypoxia response genes *fmo-2* and *acs-2* in both normoxic and hypoxic conditions, and that this regulation is dependent on *nhr-49*.

Above, we showed that autophagy induction in hypoxia is *nhr-49* dependent. To test whether *nhr-67* regulates autophagy genes in hypoxia, we examined the activity of the *lgg-1p::gfp*, *atg-2p::gfp*, and *epg-3p::gfp* reporters after *nhr-67* knockdown. Compared to the *EV(RNAi)* control, *nhr-67* knockdown unexpectedly blocked the induction of *epg-3p::gfp* by hypoxia (*Figure 4—figure supplement 1C*), but did not alter *lgg-1p::gfp* or *atg-2p::gfp* induction by hypoxia (*Figure 4A and B*, *Figure 4—figure supplement 1B*). Next, we assessed whether *nhr-67* regulates autophagosome formation in hypoxia. Following a 5 hr exposure to hypoxia, the number of LGG-1::GFP foci increased significantly and similarly in both *EV(RNAi)* control and in *nhr-67(RNAi)* animals (*Figure 4G*). This suggests that, although *nhr-67* is required for the induction of *epg-3*, its role in autophagy regulation in hypoxia is minor.

As a negative regulator of some *nhr-49*-dependent hypoxia response genes, it is possible that *nhr-67* acts upstream of *nhr-49* or directly on the promoter of *acs-2* and *fmo-2*. To determine how *nhr-67* regulates this response, we used feeding RNAi to knock down *nhr-67* and observed expression of the NHR-49::GFP translational fusion protein. Whole-animal NHR-49::GFP expression was increased in both normoxia and hypoxia following knockdown of *nhr-67*, with the highest increase observed in the intestine (*Figure 6F and G*). This suggests that *nhr-67* negatively regulates NHR-49, but in hypoxia, an increase in NHR-49 protein levels may in turn repress *nhr-67*, suggesting a negative feedback loop. The effects seen on *fmo-2* and *acs-2* expression are likely a consequence of NHR-67's effect on NHR-49.

Loss-of-function mutations in *nhr-67* cause early L1 lethality or arrest (*Fernandes and Sternberg, 2007*), so we used feeding RNAi to study *nhr-67*'s functional requirements in hypoxia. We assessed the ability of *nhr-67(RNAi)* embryos to survive hypoxia and recover, as described above. Only 58% of *nhr-67* knockdown embryos survived to at least L4 stage compared to the *EV(RNAi)* animals (82%; *Figure 6—figure supplement 1D*). Next, we used epistasis analysis to test whether *nhr-67* acts in the *nhr-49* pathway. We observed that *nhr-49(null);nhr-67(RNAi)* animals showed similar survival (17%) as do *nhr-49;EV(RNAi)* animals (29%), suggesting that these two genes act in the same pathway. In contrast, *hif-1(null);nhr-67(RNAi)* animals showed significantly reduced survival (16%) compared to *hif-1;EV(RNAi)* animals (46%), consistent with the view that *hif-1* and *nhr-49* act in separate pathways (*Figure 6—figure supplement 1D*). The majority of *nhr-67(RNAi)* animals were able to reach at least L4 stage in normoxia (98%), resembling *EV(RNAi)* animals (94%; *Figure 6—figure supplement 1E*, *Supplementary file 1*). Thus, although *nhr-67* appears to perform a negative regulatory role on the NHR-49-dependent hypoxia pathway, it, too, is functionally required for survival in hypoxia. Taken together, these data show that *nhr-67* is a functionally important negative regulator of the *nhr-49*-dependent hypoxia response, although it does not equally control all NHR-49-regulated hypoxia response genes.

## The kinase *hpk-1* positively regulates *nhr-49*-dependent hypoxia response genes and is required for survival in hypoxia

Many stress response pathways involve upstream kinases that phosphorylate transcription factors, including PMK-1 and GSK-3, which phosphorylate the oxidative stress response regulator SKN-1 (*Blackwell et al., 2015*), and AKT-1/2, which phosphorylate DAF-16 in the insulin/IGF signalling pathway (*Ogg and Ruvkun, 1998*). To identify factors acting in the *nhr-49*-dependent hypoxia response pathway, we studied kinases that we found to potentially act in the *nhr-49*-dependent oxidative stress response (Doering & Taubert, manuscript in preparation). We depleted each kinase using feeding RNAi to determine if any treatment prevented *fmo-2p::gfp* induction in hypoxia in the worm intestine. As expected, *nhr-49* RNAi diminished this intestinal fluorescence compared to the *EV(RNAi)* (*Figure 7A and B*). Of the kinases tested, RNAi knockdown of the nuclear serine/threonine

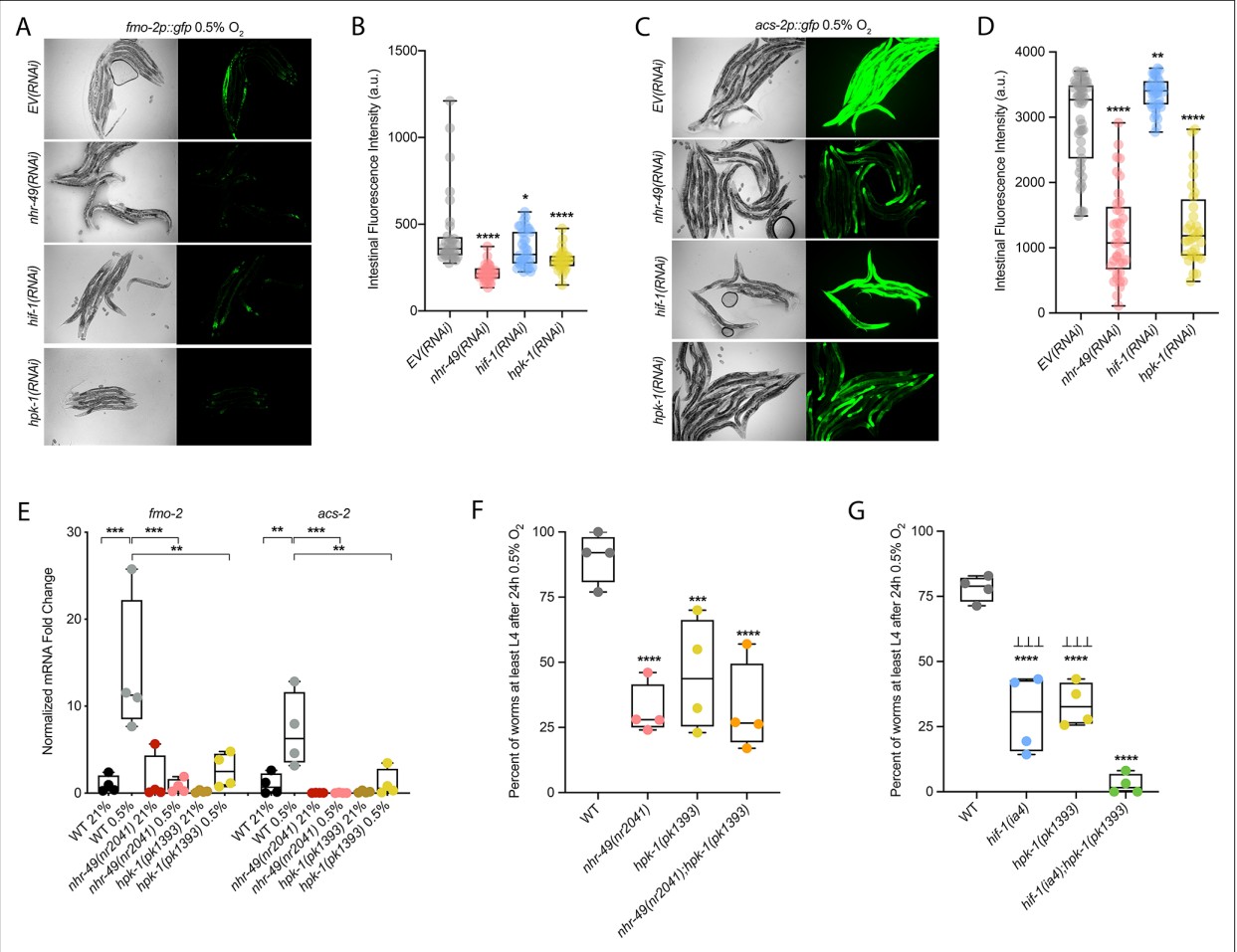

**Figure 7.** *hpk-1* is a positive regulator within the *nhr-49*-dependent hypoxia response pathway. (**A–D**) Representative micrographs and quantification of intestinal GFP levels in *fmo-2p::gfp* (**A, B**) and *acs-2p::gfp* (**C, D**) adult animals fed *EV*, *nhr-49*, *hif-1*, or *hpk-1* RNAi following 4 hr exposure to 0.5% $O_2$ and 1 hr recovery in 21% $O_2$ (three or more repeats totalling >30 individual animals per strain). *p<0.05, **p<0.01, ****p<0.0001 vs. *EV(RNAi)* (ordinary one-way ANOVA corrected for multiple comparisons using the Tukey method). (**E**) The graph shows fold changes of mRNA levels in L4 wild-type, *nhr-49(nr2041)*, and *hpk-1(pk1393)* animals exposed to 0.5% $O_2$ for 3 hr (n = 4). **p<0.01, ***p<0.001 (two-way ANOVA corrected for multiple comparisons using the Tukey method). (**F**) The graph shows the average population survival of wild-type, *nhr-49(nr2041)*, *hpk-1(pk1393)*, and *nhr-49(nr2041);hpk-1(pk1393)* embryos following 24 hr exposure to 0.5% $O_2$, then allowed to recover at 21% $O_2$ for 65 hr, and counted as the ability to reach at least L4 stage (four repeats totalling >100 individual animals per strain). ***p<0.001, ****p<0.0001 vs. wild-type animals. Comparison of single mutants to *nhr-49(nr2041);hpk-1(pk1393)* not significant (ordinary one-way ANOVA corrected for multiple comparisons using the Tukey method). (**G**) The graph shows the average population survival of wild-type, *hif-1(ia4)*, *hpk-1(pk1393)*, and *hif-1(ia4);hpk-1(pk1393)* embryos following 24 hr exposure to 0.5% $O_2$, then allowed to recover at 21% $O_2$ for 65 hr, and counted as the ability to reach at least L4 stage (four repeats totalling >100 individual animals per strain). ****p<0.0001 vs. wild-type animals, ⊥⊥⊥p<0.001 vs. *hif-1(ia4);hpk-1(pk1393)* (ordinary one-way ANOVA corrected for multiple comparisons using the Tukey method). n.s.: not significant; WT: wild-type. See *Source data 1* for (**B, D, E–G**).

The online version of this article includes the following figure supplement(s) for figure 7:

**Figure supplement 1.** *hpk-1* is required for *fmo-2* induction, and *hpk-1* mutants do not display functional defects in normoxia.

kinase *hpk-1* significantly decreased intestinal *fmo-2p::gfp* expression following hypoxia exposure (**Figure 7A and B**), phenocopying *nhr-49* knockdown. Knockdown of *hpk-1* also significantly reduced intestinal expression of the *acs-2p::gfp* reporter in hypoxia (**Figure 7C and D**) and reduced expression of *fmo-2p::gfp* in the *nhr-49(et13)* background in normoxia (**Figure 7—figure supplement 1A and B**). In comparison, *hif-1* RNAi significantly decreased the expression of the *fmo-2p::gfp* reporter in hypoxia (**Figure 7A–B**) but did not alter it in the *nhr-49(et13)* background (**Figure 7—figure supplement 1A, B**), and actually increased expression of the *acs-2p::gfp* reporter in hypoxia (**Figure 7C and D**). We corroborated the *hpk-1* data using qRT-PCR in wild-type animals and in a *hpk-1(pk1393)* mutant. The *pk1393* allele deletes the majority of the kinase domain of *hpk-1* and is a predicted

molecular null allele (*Raich et al., 2003*). In hypoxia, the expression of both *acs-2* and *fmo-2* was significantly reduced by loss of *hpk-1*, phenocopying loss of *nhr-49* (*Figure 7E*). Together, these data suggest that, like *nhr-49*, *hpk-1* is required for upregulation of *fmo-2* and *acs-2* in response to hypoxia.

*hpk-1* regulates autophagy in response to dietary restriction in *C. elegans* (*Das et al., 2017*). To determine if *hpk-1* is involved in the regulation of autophagy in response to hypoxia, like *nhr-49*, we examined the expression of the *lgg-1p::gfp*, *atg-2p::gfp*, and *epg-3p::gfp* transcriptional reporters. Similar to *nhr-49*, induction of all three autophagy genes in hypoxia required *hpk-1* (*Figure 4A and B*, *Figure 4—figure supplement 1B and C*). Next, we assessed whether *hpk-1* is necessary for autophagosome formation in hypoxia. Following a 5 hr exposure to hypoxia, the number of LGG-1::GFP foci was not changed compared to the *hpk-1(RNAi)* normoxia control (*Figure 4G*). This shows that, like *nhr-49*, *hpk-1* is required for the induction of autophagy genes and autophagosome formation in hypoxia.

To determine if *hpk-1* is functionally required for animal survival in hypoxia, we assessed the ability of *hpk-1* mutant embryos to survive hypoxia. Similar to *nhr-49* mutant animals, only 45% of *hpk-1* mutant embryos developed to L4 (wild-type animals 92%; *Figure 7F*). We used epistasis analysis to test the hypothesis that *hpk-1* acts in the *nhr-49* pathway to coordinate a transcriptional response to hypoxia. We observed that the *nhr-49;hpk-1* double null mutant showed similar survival (26%) to each of the single null mutants, suggesting that these two genes act in the same hypoxia response pathway (*Figure 7F*). In contrast, the *hif-1;hpk-1* double null mutant was significantly impaired (<2%) compared to each of the single null mutants alone, consistent with the view that these two genes act in separate response pathways (*Figure 7G*). Each mutant showed normal development from embryo to L4 in normoxia, indicating that the phenotypes observed were specifically due to the requirement of these genes in hypoxia survival (*Figure 7—figure supplement 1C and D*, *Supplementary file 1*). Taken together, these experiments show that *hpk-1* is required for embryo survival in hypoxia, consistent with it playing a role as an activator of the *nhr-49*-dependent response pathway.

## NHR-49 is regulated post-transcriptionally in hypoxia in an *hpk-1*-dependent fashion

To test our hypothesis that HPK-1 activates NHR-49 in hypoxia, we examined whether NHR-49 is induced by hypoxia and whether *hpk-1* is involved in this regulation. NHR-49 and HPK-1 protein levels

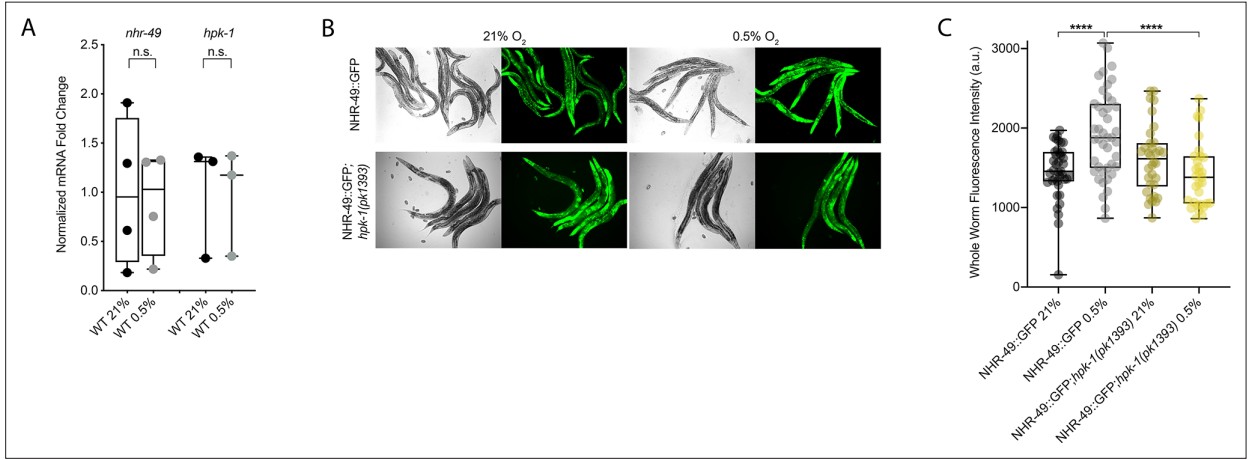

**Figure 8.** NHR-49 is induced in hypoxia in an *hpk-1*-dependent fashion. (**A**) The graph shows the average fold changes of mRNA levels (relative to unexposed wild type) in L4 wild-type animals exposed to 0.5% $O_2$ for 3 hr (n = 3 or 4; ordinary one-way ANOVA corrected for multiple comparisons using the Tukey method). (**B**) Representative micrographs show *nhr-49p::nhr-49::gfp* and *nhr-49p::nhr-49::gfp;hpk-1(pk1393)* adult animals following 4 hr exposure to 0.5% $O_2$ and 1 hr recovery in 21% $O_2$. (**C**) The graph shows the quantification of whole-worm GFP levels in *nhr-49p::nhr-49::gfp* and *nhr-49p::nhr-49::gfp;hpk-1(pk1393)* animals following 4 hr exposure to 0.5% $O_2$ and 1 hr recovery in 21% $O_2$ (three repeats totalling >30 individual animals per strain). ****$p<0.0001$ (two-way ANOVA corrected for multiple comparisons using the Tukey method). n.s.: not significant; WT: wild-type. See *Source data 1* for (**A, C**).

The online version of this article includes the following figure supplement(s) for figure 8:

**Figure supplement 1.** *hpk-1* is not transcriptionally regulated in hypoxia.

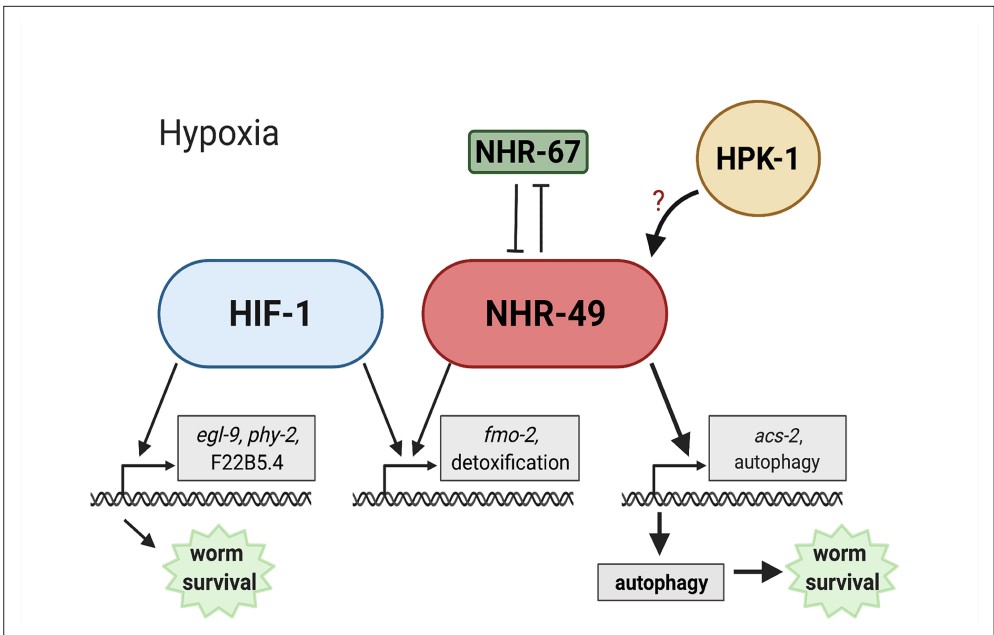

**Figure 9.** Model of the new NHR-49 hypoxia response pathway and its interaction with HIF-1 signalling. The proposed model of how NHR-49 regulates a new hypoxia response parallel to HIF-1. During normoxia, the transcription factor NHR-67 negatively regulates NHR-49. However, during hypoxia, NHR-49 represses *nhr-67*, and the kinase HPK-1 positively regulates NHR-49, possibly directly or indirectly. This allows NHR-49 to activate its downstream hypoxia response target genes, including *fmo-2*, *acs-2*, and autophagy genes, whose induction is required for worm survival to hypoxia. (Figure created with https://biorender.com/, Toronto, ON, Canada).

are increased in response to tert-butyl hydroperoxide and/or heat shock, respectively, but mRNA levels remain unchanged (**Das et al., 2017**; **Goh et al., 2018**). Similarly, we observed that *nhr-49* and *hpk-1* mRNA levels were not increased upon exposure to hypoxia (**Figure 8A**). Consistent with this, a transcriptional reporter of the *hpk-1* promoter fused to GFP (**Das et al., 2017**) was also not induced following hypoxia exposure (**Figure 8—figure supplement 1A and B**). These data show that the transcription of neither *nhr-49* nor *hpk-1* is induced in hypoxia.

We considered the possibility that NHR-49 may be regulated post-transcriptionally. To assess NHR-49 protein levels, we again used the translational NHR-49::GFP reporter to measure the expression of the fusion protein in response to hypoxia. As described above, the whole-worm NHR-49::GFP signal was modestly, but significantly, elevated upon exposure to hypoxia (**Figure 8B and C**). Interestingly, although *hpk-1* null mutation had no effect on NHR-49::GFP levels in normoxia, it abrogated the upregulation of the NHR-49::GFP signal by hypoxia (**Figure 8B and C**). Higher magnification images showed that NHR-49 is upregulated in the head, intestine, and hypodermal seam cells during hypoxia, and loss of *hpk-1* abrogated NHR-49 induction in all three tissues (**Figure 2D**, **Figure 2—figure supplement 1E and F**). This suggests that NHR-49 is regulated post-translationally in response to hypoxia, and that *hpk-1* may be involved in this regulation. Taken together, these data show that *hpk-1* is a functionally important upstream positive regulator of the *nhr-49*-dependent hypoxia response.

## Discussion

Animals, tissues, and cells must be able to rapidly, flexibly, and reversibly adapt to a plethora of stresses. Past studies have identified many stress response factors, often termed master regulators. However, recent studies indicate that stress response regulation requires the intricate interactions of multiple factors as part of networks that provide regulatory redundancy and flexibility. NHR-49 is a transcription factor that promotes longevity and development by regulating lipid metabolism and various stress responses (**Chamoli et al., 2014**; **Dasgupta et al., 2020**; **Goh et al., 2018**; **Naim et al., 2021**; **Wani et al., 2021**). Our data show that NHR-49 coordinates a part of the transcriptional response to hypoxia. The NHR-49 pathway operates in parallel to the canonical HIF-1 hypoxia response pathway.

Besides *nhr-49*, this pathway includes *nhr-67* and *hpk-1*. The former interacts with NHR-49 (*Reece-Hoyes et al., 2013*), potentially forming a regulatory NHR-NHR heterodimer that modulates NHR-49 activity. During normoxia, *nhr-67* acts to repress NHR-49; however, during hypoxia, an increase in NHR-49 protein levels in turn represses *nhr-67* levels, forming a feedback loop that may serve to reinforce NHR-49 activity. In contrast to *nhr-67*, the upstream kinase HPK-1 positively regulates at least part of the NHR-49-dependent hypoxia response, either directly or indirectly, as it is required to survive hypoxia and to activate the NHR-49-regulated hypoxia response genes, including *fmo-2*, *acs-2*, and autophagy genes. Downstream, NHR-49 and HPK-1 induce autophagy, which is essential to promote hypoxia survival. Collectively, our experiments delineate a hypoxia response pathway that contains distinct upstream and downstream components and is just as essential for hypoxia survival as the parallel *hif-1* pathway (*Figure 9*).

## NHR-49 controls a novel hypoxia response pathway that is parallel to canonical HIF signalling

*nhr-49* is required to induce *fmo-2* in various stresses and infection models (*Chamoli et al., 2014*; *Dasgupta et al., 2020*; *Goh et al., 2018*; *Naim et al., 2021*; *Wani et al., 2021*). Similarly, HIF-1 regulates *fmo-2* in several *C. elegans* longevity paradigms (*Leiser et al., 2015*), and *fmo-2* is induced in hypoxia, specifically 0.1% $O_2$ exposure, in a *hif-1*-dependent manner (*Leiser et al., 2015*; *Shen et al., 2005*). This raised the possibility that *hif-1* also promoted *fmo-2* expression in hypoxia (0.5% $O_2$) in L4 or older worms, and, more generally, that *nhr-49* might act through *hif-1* in the hypoxia response. However, several lines of evidence support a model whereby HIF-1 and NHR-49 are core components of parallel signalling networks (*Figure 9*). First, *hif-1* and *nhr-49* interact genetically in hypoxia survival experiments, suggesting that they work in parallel genetic pathways (*Figure 2A, B and E*). Second, our transcriptome analysis identified sets of genes that are regulated exclusively by HIF-1 or NHR-49 (*Figure 3A and B*). Third, the kinase *hpk-1* and the transcription factor *nhr-67* show synthetic genetic interaction with *hif-1*, but not with *nhr-49* (*Figure 7F and G*, *Figure 6—figure supplement 1D*). In support of our study, a recent publication (*Vozdek et al., 2018*) showed that *nhr-49* is required to induce the *hif-1*-independent hypoxia response gene *comt-5* both in 0.5% $O_2$ and in a strain mutant for the kinase *hir-1*. In hypoxia, HIR-1 coordinates remodelling of the extracellular matrix independently of HIF-1 (*Vozdek et al., 2018*). Thus, although our RNA-seq results did not identify *comt-5* as a target of NHR-49 in hypoxia, this study supports the idea of a *nhr-49* hypoxia response pathway that acts in parallel with *hif-1*.

## Homeodomain-interacting protein kinases in hypoxia

Our efforts to map additional components of the NHR-49 hypoxia response pathway, especially factors acting in concert with NHR-49, revealed HPK-1 (*Figure 9*). Homeodomain-interacting protein kinases (HIPKs) are a family of nuclear serine/threonine kinase that can phosphorylate transcription factors (*Rinaldo et al., 2007*; *Rinaldo et al., 2008*). The worm's only HIPK orthologue, *hpk-1*, regulates development and the response to DNA damage, heat shock, and dietary restriction (*Berber et al., 2013*; *Berber et al., 2016*; *Das et al., 2017*; *Rinaldo et al., 2007*). Notably, *hpk-1* regulates autophagy in response to dietary restriction as it is necessary to induce autophagosome formation and autophagy gene expression (*Das et al., 2017*). Here, we show that *hpk-1* is an upstream regulator of the *nhr-49*-dependent hypoxia response pathway. Our data suggest that HPK-1 promotes the accumulation of NHR-49 protein in hypoxia, leading to induction of NHR-49-dependent hypoxia response genes. This includes the induction of autophagy genes and autophagosome formation in hypoxia (*Figure 4A, B and G*, *Figure 4—figure supplement 1B and C*). In line with our model, mammalian HIPK2 is induced in and required to protect cardiomyocytes from hypoxia/reoxygenation induced injury (*Dang et al., 2020*). In contrast, in breast cancer cells, HIPK2 is degraded during periods of low oxygen via association with the E3 ubiquitin ligase SIAH2 (*Calzado et al., 2009*); this degradation of HIPK2 is necessary as the protein normally represses the expression of HIF-1α by binding at its promoter (*Nardinocchi et al., 2009*). Thus, protecting cells from hypoxic injury may be a conserved, albeit cell-type-specific, role of HIPKs. Future experiments may reveal how HPK-1 regulates NHR-49, perhaps by examining direct phosphorylation and activation of the NHR-49 protein by HPK-1.

## Paradoxical regulation of the β-oxidation gene *acs-2* by hypoxia

Mitochondria consume cellular oxygen to produce energy and thus must adapt to limited oxygen availability. In particular, mitochondrial β-oxidation, the consumption of oxygen to catabolize fatty acids for energy production, is repressed in hypoxia in favour of anaerobic respiration. For example, the heart and skeletal muscle of mice and rats show decreased expression of key β-oxidation enzymes in acute hypoxia (*Kennedy et al., 2001*; *Morash et al., 2013*). In *C. elegans*, the acyl-CoA synthetase *acs-2* is part of the mitochondrial β-oxidation pathway, where it functions in the first step to activate fatty acids. NHR-49 activates *acs-2* expression during starvation, when β-oxidation is induced (*Van Gilst et al., 2005b*). Considering this, *acs-2* expression would be expected to be downregulated in hypoxia due to reduced β-oxidation. Paradoxically, however, we found that *acs-2* is strongly induced in hypoxia and that this regulation depends on *nhr-49* (*Figure 3C*, *Figure 3—figure supplement 2A–C*). Examination of other fatty acid β-oxidation enzymes in our RNA-seq data showed that *acs-2* is the only enzyme induced. This suggests that, during hypoxia, ACS-2 is not feeding its product fatty acyl-CoA into the β-oxidation cycle, but perhaps produces fatty acyl-CoA for anabolic functions needed for survival in or recovery from low oxygen, such as phospholipid or triglyceride synthesis (reviewed in *Tang et al., 2018*). Similar functions have been observed in human macrophages, which, during hypoxia, decrease β-oxidation but increase triglyceride synthesis (*Boström et al., 2006*).

In line with the repression of β-oxidation in hypoxia (*Boström et al., 2006*; *Kennedy et al., 2001*; *Morash et al., 2013*), there is evidence supporting a HIF-dependent downregulation of the mammalian NHR-49 homolog PPARα, which promotes β-oxidation (*Atherton et al., 2008*). For example, in human hepatocytes and mouse liver sections, HIF-2α accumulation in hypoxia directly suppresses PPARα expression (*Chen et al., 2019*). Additionally, HIF-1α suppresses PPARα protein and mRNA levels during hypoxia in intestinal epithelial cells, and the *PPARA* promoter contains a HIF-1α DNA-binding consensus motif, suggesting direct control of *PPARA* by HIF transcription factors (*Narravula and Colgan, 2001*).

Some evidence suggests alternative actions of PPARα. Knockdown of PPARα attenuates the ability of Phd1 (a homolog of *C. elegans egl-9*) knockout myofibers to successfully tolerate hypoxia (*Aragonés et al., 2008*), suggesting that PPARα is an important regulator of the hypoxia response downstream of Phd1. Along these lines, PPARα protein levels increase in the muscle of Phd1 knockout mice (*Aragonés et al., 2008*) and following hypoxic exposure in mouse hearts (*Morash et al., 2013*). Similarly, we show that NHR-49 protein levels increase in response to hypoxia (*Figures 2D, 8B and C*, *Figure 2—figure supplement 1C–F*), and that NHR-49 is a vital regulator of a hypoxia response that works in parallel with HIF-1. Together, these data suggest that, similar to evidence from studies in mammalian systems, NHR-49 levels are increased and required in hypoxia, and may be regulating *acs-2* for functions other than fatty acid β-oxidation.

## NHR-49 promotes autophagy activation to achieve hypoxia survival

During stress, damaged cellular components can be cleared or recycled via autophagy, a key process regulated by *nhr-49* in hypoxia (*Figures 3D and 4*). Autophagy is part of an adaptive response to hypoxia. During periods of low oxygen, cells switch from aerobic mitochondrial respiration to anaerobic glycolysis. To meet this increased glycolytic demand, the autophagy machinery promotes the activity and cell surface expression of the glucose transporter GLUT1 to increase cellular glucose uptake (*Roy et al., 2017*). In addition, hypoxia causes improper protein folding in the endoplasmic reticulum (ER), activating the unfolded protein response (UPR). Although the exact mechanism is unknown, it is thought that autophagy and the UPR are activated simultaneously during stress to restore homeostasis, and that autophagy can assist in alleviating ER stress when the UPR is disrupted or overwhelmed (reviewed in *Yan et al., 2015*).

In mammals, PPARα activates autophagy in response to various stresses, including in neurons to clear Aβ in Alzheimer's disease (*Luo et al., 2020*), and in the liver during inflammation (*Jiao et al., 2014*) and starvation (*Lee et al., 2014*). Proper regulation of autophagy is also a requirement in hypoxic conditions. Knockdown or genetic mutation of various *C. elegans* autophagy genes showed that they are required for worm survival when worms experience anoxia and elevated temperatures combined (*Samokhvalov et al., 2008*). Similarly, Zhang et al. found that mitochondrial autophagy (mitophagy) is induced by hypoxia in mouse embryo fibroblasts. This process requires the expression of BNIP3 (Bcl-2/E1B 19 kDa-interacting protein 3), an autophagy regulator, which is induced in

a HIF-1-dependent manner (*Zhang et al., 2008*). In agreement with this, our RNA-seq data showed a 3.8-fold induction of the *C. elegans* BNIP3 homolog *dct-1* in hypoxia; however, this induction was dependent on neither *nhr-49* nor *hif-1*. The above study also found that the autophagy genes Beclin-1 and Atg5 are induced and required for cell survival in hypoxia (*Zhang et al., 2008*). Here, we show for the first time that autophagy is both induced and required for *C. elegans* adaptation and survival to 0.5% $O_2$. The *C. elegans* orthologue of Beclin-1, *bec-1*, and the worm *lgg-2*, *epg-6*, *epg-3*, *atg-7*, and *atg-10* genes, which are involved in the completion of the autophagosome along with *atg-5/Atg5*, are required for worm embryo survival to hypoxia in an *nhr-49*-dependent manner (*Figure 4E and F*, *Figure 4—figure supplement 1F and G*). In addition, we show that both *nhr-49* and *hpk-1* are required to induce the expression of autophagy genes and autophagosome formation during hypoxia, processes that are independent of *hif-1*. In agreement with our findings, Valko et al. recently reported that the formation of autophagosomes by hypoxia is independent of *hif-1/sima* in *Drosophila melanogaster* (*Valko et al., 2021*).

## Cell non-autonomous functions of NHR-49 in hypoxia

Cell non-autonomous regulation occurs in many pathways in *C. elegans*. For example, HIF-1 acts in neurons to induce *fmo-2* expression in the intestine to promote longevity (*Leiser et al., 2015*). NHR-49 is expressed in the intestine, neurons, muscle, and hypodermis (*Ratnappan et al., 2014*). Re-expression of *nhr-49* in any one of these tissues is sufficient to enhance worm survival upon infection with the pathogens *S. aureus* (*Wani et al., 2021*) and to promote longevity in germline-less animals (*Naim et al., 2021*), but NHR-49 acts only in neurons to promote survival upon infection by *P. aeruginosa* (*Naim et al., 2021*). We thus aimed to identify the key tissue wherein NHR-49 promotes hypoxia survival. Surprisingly, we found that *nhr-49* expression in any of the intestine, neurons, hypodermis, or body wall muscle is sufficient for whole-animal survival to hypoxia (*Figure 5B*), suggesting that NHR-49 can act in a cell non-autonomous fashion to execute its effects. Possibly, a signalling molecule whose synthesis is promoted by NHR-49 activity in any tissue promotes organismal hypoxia adaptation. It is also possible that ectopic overexpression of NHR-49 shifts metabolism in the tissue wherein it is expressed, releasing metabolites that promote organismal hypoxia adaptation.

In sum, we show here that NHR-49 regulates a novel hypoxia response pathway parallel to HIF-1 and controls an important transcriptional response for worm survival in hypoxia. If the mammalian NHR-49 homologs PPARα and HNF4 play similar roles in the cellular response to hypoxia, our discovery could lead to the identification and development of new targets for drugs and therapies for diseases exhibiting hypoxic conditions.

# Materials and methods

## Key resources table

| Reagent type (species) or resource | Designation | Source or reference | Identifiers | Additional information |
|---|---|---|---|---|
| Strain, strain background (*Escherichia coli*) | OP50 | Caenorhabditis Genetics Center (CGC) | | |
| Strain, strain background (*E. coli*) | HT115 | Caenorhabditis Genetics Center (CGC) | | |
| Genetic reagent (*Caenorhabditis elegans*) | N2 | Caenorhabditis Genetics Center (CGC) (*Brenner, 1974*) | | |
| Genetic reagent (*C. elegans*) | *nhr-49(nr2041) I* | PMID:15719061 (*Van Gilst et al., 2005a*) | STE68; RRID:WB-STRAIN:WBStrain00034504 | |
| Genetic reagent (*C. elegans*) | *eavEx20[fmo-2p::gfp+rol-6(su1006)]* | PMID:29508513 (*Goh et al., 2018*) | VE40 | |
| Genetic reagent (*C. elegans*) | *nhr-49(nr2041) I; eavEx20[fmo-2p::gfp+rol-6(su1006)]* | This study | STE129 | |

*Continued on next page*

*Continued*

| Reagent type (species) or resource | Designation | Source or reference | Identifiers | Additional information |
|---|---|---|---|---|
| Genetic reagent (*C. elegans*) | *hif-1(ia4) V* | PMID:11427734 (*Jiang et al., 2001*) | ZG31; RRID:WB-STRAIN:WBStrain00040824 | |
| Genetic reagent (*C. elegans*) | *nhr-49(nr2041) I; hif-1(ia4) V* | This study | STE130 | |
| Genetic reagent (*C. elegans*) | *fmo-2(ok2147) IV* | PMID:26586189 (*Leiser et al., 2015*) | VC1668; RRID:WB-STRAIN:WBStrain00036780 | |
| Genetic reagent (*C. elegans*) | *acs-2(ok2457) V* | PMID:21704635 (*Zhang et al., 2011*) | RB1899 | |
| Genetic reagent (*C. elegans*) | *fmo-2(ok2147) IV; acs-2(ok2457) V* | This study | STE131 | |
| Genetic reagent (*C. elegans*) | *nhr-49(et13) I* | PMID:27618178 (*Lee et al., 2016*) | STE110 | |
| Genetic reagent (*C. elegans*) | *nhr-49(nr2041) I;glmEx5 [nhr-49p::nhr-49::gfp+myo-2p::mCherry]* | PMID:34156142 (*Naim et al., 2021*) | AGP33a | |
| Genetic reagent (*C. elegans*) | *nhr-49(nr2041) I; glmEx9 [gly-19p::nhr-49::gfp+myo-2p::mCherry]* | PMID:34156142 (*Naim et al., 2021*) | AGP65 | |
| Genetic reagent (*C. elegans*) | *nhr-49(nr2041)I; glmEx11 [col-12p::nhr-49::gfp+myo-2p::mCherry]* | PMID:34156142 (*Naim et al., 2021*) | AGP53 | |
| Genetic reagent (*C. elegans*) | *nhr-49(nr2041)I; glmEx13 [rgef-1p::nhr-49::gfp+myo-2p::mCherry]* | PMID:34156142 (*Naim et al., 2021*) | AGP51 | |
| Genetic reagent (*C. elegans*) | *nhr-49(nr2041)I; glmEx8 [myo-3p::nhr-49::gfp+myo-2p::mCherry]* | PMID:34156142 (*Naim et al., 2021*) | AGP63 | |
| Genetic reagent (*C. elegans*) | *wbmEx57 [acs-2p::gfp+rol-6(su1006)]* | PMID:25723162 (*Burkewitz et al., 2015*) | WBM170 | |
| Genetic reagent (*C. elegans*) | *nhr-49(nr2041) I; wbmEx57 [acs-2p::gfp+rol-6(su1006)]* | PMID:25723162 (*Burkewitz et al., 2015*) | WBM169 | |
| Genetic reagent (*C. elegans*) | *glmEx5 (nhr-49p::nhr-49::gfp+myo-2p::mCherry)* | PMID:25474470 (*Ratnappan et al., 2014*) | AGP25f | |
| Genetic reagent (*C. elegans*) | *hif-1(ia4) V; glmEx5 (nhr-49p::nhr-49::gfp+myo-2p::mCherry)* | This study | STE140 | |
| Genetic reagent (*C. elegans*) | *hpk-1(pk1393) X; glmEx5 (nhr-49p::nhr-49::gfp+myo-2p::mCherry)* | This study | STE142 | |
| Genetic reagent (*C. elegans*) | *hpk-1(pk1393) X* | PMID:12618396 (*Raich et al., 2003*) | EK273; RRID:WB-STRAIN:WBStrain00007138 | |
| Genetic reagent (*C. elegans*) | *nhr-49(nr2041) I; hpk-1(pk1393) X* | This study | STE132 | |
| Genetic reagent (*C. elegans*) | *hif-1(ia4) V; hpk-1(pk1393) X* | This study | STE133 | |
| Genetic reagent (*C. elegans*) | *nhr-49(et13) I; eavEx20[fmo-2p::gfp+rol-6(su1006)]* | PMID:29508513 (*Goh et al., 2018*) | STE117 | |
| Genetic reagent (*C. elegans*) | *artEx12 [hpk-1p::gfp+rol-6(su1006)]* | PMID:29036198 (*Das et al., 2017*) | AVS394 | |

*Continued on next page*

*Continued*

| Reagent type (species) or resource | Designation | Source or reference | Identifiers | Additional information |
|---|---|---|---|---|
| Genetic reagent (*C. elegans*) | *dpy-5(e907) I; sEx14068 [rCes atg-2::GFP+pCeh361]* | PMID:15338614 (*McKay et al., 2003*) | BC14068 | |
| Genetic reagent (*C. elegans*) | *dpy-5(e907) I; sEx13567 [rCes lgg-1::GFP+pCeh361]* | PMID:15338614 (*McKay et al., 2003*) | BC13567 | |
| Genetic reagent (*C. elegans*) | *dpy-5(e907) I; sEx10273 [rCes epg-3::GFP+pCeh361]* | PMID:15338614 (*McKay et al., 2003*) | BC10273 | |
| Genetic reagent (*C. elegans*) | *adIs2122 [lgg-1p::GFP::lgg-1 + rol-6(su1006)]* | PMID:17785524 (*Kang et al., 2007*) | DA2123 | |
| Genetic reagent (*C. elegans*) | *nhr-49(nr2041) I; adIs2122 [lgg-1p::GFP::lgg-1 + rol-6(su1006)]* | This study | STE143 | |
| Genetic reagent (*C. elegans*) | *hif-1(ia4) V; adIs2122 [lgg-1p::GFP::lgg-1 + rol-6(su1006)]* | This study | STE144 | |
| Genetic reagent (*C. elegans*) | *lgg-2(tm5755) IV* | PMID:24374177 (*Manil-Ségalen et al., 2014*) | RD220 | |
| Genetic reagent (*C. elegans*) | *nhr-49(nr2041) I; lgg-2(tm5755) IV* | This study | STE145 | |
| Genetic reagent (*C. elegans*) | *epg-6(tm8366) III* | This study, non-outcrossed mutant obtained from NBRP; PMID:19934255 | STE147 | |
| Genetic reagent (*C. elegans*) | *nhr-49(nr2041) I; epg-6(tm8366) III* | This study | STE146 | |
| Sequence-based reagent | Source BioScience | PMID:11099033 | RNAi clones | |
| Sequence-based reagent | fmo-2_F | This paper | qPCR primer | GGAACAAGCGTGTTGCTGT |
| Sequence-based reagent | fmo-2_R | This paper | qPCR primer | GCCATAGAGAAGACCATGTCG |
| Sequence-based reagent | acs-2_F | This paper | qPCR primer | AGTGAGACTTGACAGTTCCG |
| Sequence-based reagent | acs-2_R | This paper | qPCR primer | CTTGTAAGAGAGGAATGGCTC |
| Sequence-based reagent | nhr-49_F | This paper | qPCR primer | TCCGAGTTCATTCTCGACG |
| Sequence-based reagent | nhr-49_R | This paper | qPCR primer | GGATGAATTGCCAATGGAGC |
| Sequence-based reagent | hpk-1_F | This paper | qPCR primer | TGTCAAAGTGAAGCCGCTGG |
| Sequence-based reagent | hpk-1_R | This paper | qPCR primer | CGGCGCCAGTTCGTGTAGTA |
| Sequence-based reagent | nhr-67_F | This paper | qPCR primer | GAGGATGATGCGACGAGTAG |
| Sequence-based reagent | nhr-67_R | This paper | qPCR primer | TGGTCTTGAAGAGGAAGGGGA |
| Sequence-based reagent | act-1_F | This paper | qPCR primer | GCTGGACGTGATCTTACTGATTACC |
| Sequence-based reagent | act-1_R | This paper | qPCR primer | GTAGCAGAGCTTCTCCTTGATGTC |
| Sequence-based reagent | tba-1_F | This paper | qPCR primer | GTACACTCCACTGATCTCTGCTGACAAG |
| Sequence-based reagent | tba-1_R | This paper | qPCR primer | CTCTGTACAAGAGGCAAACAGCCATG |
| Sequence-based reagent | ubc-2_F | This paper | qPCR primer | AGGGAGGTGTCTTCTTCCTCAC |
| Sequence-based reagent | ubc-2_R | This paper | qPCR primer | CGGATTTGGATCACAGAGCAGC |
| Sequence-based reagent | oac-14_F | This paper | qPCR primer | TTCCAGCGACTTTTCTTTCG |
| Sequence-based reagent | oac-14_R | This paper | qPCR primer | CCCAGGATTGCTTCAATCAG |
| Sequence-based reagent | cyp-13A11_F | This paper | qPCR primer | ACACGTGGACACTTCACTATG |

*Continued on next page*

*Continued*

| Reagent type (species) or resource | Designation | Source or reference | Identifiers | Additional information |
|---|---|---|---|---|
| Sequence-based reagent | cyp-13A11_R | This paper | qPCR primer | TTCCGATACACTGTCGAGGTC |
| Sequence-based reagent | cyp-25A3_F | This paper | qPCR primer | agaatcgttgctccaaaacac |
| Sequence-based reagent | cyp-25A3_R | This paper | qPCR primer | ttcaaaatctccaggaacagg |
| Sequence-based reagent | ugt-20_F | This paper | qPCR primer | CCGACAAATCCCAGAGAGACA |
| Sequence-based reagent | ugt-20_R | This paper | qPCR primer | TGTCCAAAAAGAAGTACTCAACG |
| Sequence-based reagent | atg-2_F | This paper | qPCR primer | AGATGTCCGCCATAGTCTGC |
| Sequence-based reagent | atg-2_R | This paper | qPCR primer | TCTTCCTGAGCAGCGAGTTC |
| Sequence-based reagent | epg-9_F | This paper | qPCR primer | CGACGAAAACCGAGATTCCC |
| Sequence-based reagent | epg-9_R | This paper | qPCR primer | TGAGCCAGCGATTGTTTGTG |
| Sequence-based reagent | lgg-2_F | This paper | qPCR primer | GCAGTTTACCACTTATGGATCGC |
| Sequence-based reagent | lgg-2_R | This paper | qPCR primer | CGTTCATTGACGAGCAGGAAG |
| Sequence-based reagent | atg-13_F | This paper | qPCR primer | AAGCAGCTGAAAACTGCTCC |
| Sequence-based reagent | atg-13_R | This paper | qPCR primer | CGGAGAACGAATTGACGTGTT |
| Sequence-based reagent | Random primers | Invitrogen | 48190-011 | |
| Sequence-based reagent | dNTPs | Fermentas | R0186 | |
| Chemical compound, drug | Carbenicillin | BioBasic | CDJ469 | |
| Chemical compound, drug | IPTG | Santa Cruz | sc-202185B | CAS 367-93-1 |
| Chemical compound, drug | Tetracycline | BioBasic | TB0504 | |
| Chemical compound, drug | RNAseOUT | Invitrogen | 10777-019 | |
| Chemical compound, drug | Fast SYBR Master Mix | Life Technologies | 4385612 | |
| Chemical compound, drug | Levamisole | Sigma | L9756 | |
| Chemical compound, drug | $H_2S$ | AirGas, Seattle, WA | X02NI99CP581327 | |
| Chemical compound, drug | 5000 ppm $O_2$ balanced with $N_2$ | Praxair Canada | NI OX5000C–T | |
| Software, algorithm | ImageJ | PMID:22930834 | | https://imagej.nih.gov/ij/index.html |
| Software, algorithm | Trimmomatic version 0.36 | PMID:24695404 | RRID:SCR_011848 | |
| Software, algorithm | Salmon version 0.9.1 | PMID:28263959 | RRID:SCR_017036 | https://combine-lab.github.io/salmon/ |
| Software, algorithm | tximport | PMID:26925227 | RRID:SCR_016752 | https://github.com/mikelove/tximport |
| Software, algorithm | edgeR | PMID:19910308 | RRID:SCR_012802 | http://bioconductor.org/packages/edgeR/ |
| Software, algorithm | eVITTA | PMID:34019643 | | https://tau.cmmt.ubc.ca/eVITTA/ |

## Nematode strains and growth conditions

We cultured *C. elegans* strains using standard techniques on nematode growth media (NGM) plates. To avoid background effects, each mutant was crossed into our lab N2 strain; original mutants were backcrossed to N2 at least six times, except *lgg-2* and *epg-6* mutants, which were backcrossed four times. *Escherichia coli* OP50 was the food source in all experiments except for RNAi experiments, where we used *E. coli* HT115. All experiments were carried out at 20°C. Worm strains used in this study are listed in the Key resources table. For synchronized worm growths, we isolated embryos by standard sodium hypochlorite treatment. Isolated embryos were allowed to hatch overnight on unseeded NGM plates until the population reached a synchronized halted development at L1 stage

via short-term fasting (12–24 hr). Synchronized L1 stage larvae were then transferred to OP50 seeded plates and grown to the desired stage.

## Feeding RNAi

RNAi was performed on NGM plates supplemented with 25 µg/ml carbenicillin (BioBasic CDJ469), 1 mM IPTG (Santa Cruz CAS 367-93-1), and 12.5 µg/ml tetracycline (BioBasic TB0504; NGM-RNAi plates), and seeded with appropriate HT115 RNAi bacteria. The RNAi clones were from the Ahringer library (Source BioScience) and were sequenced prior to use.

## RNA isolation and qRT-PCR analysis

Synchronized L1 worms were allowed to grow on OP50 plates for 48 hr to L4 stage, then either kept in 21% $O_2$ or transferred to 0.5% $O_2$ for 3 hr and rapidly harvested. RNA isolation was performed as previously described (*Goh et al., 2014*). 2 µg total RNA was used to generate cDNA with Superscript II reverse transcriptase (Invitrogen 18064-014), random primers (Invitrogen 48190-011), dNTPs (Fermentas R0186), and RNAseOUT (Invitrogen 10777-019). Quantitative PCR was performed in 10 µl reactions using Fast SYBR Master Mix (Life Technologies 4385612), 1:10 diluted cDNA, and 5 µM primer, and analysed with an Applied Biosystems StepOnePlus machine. We analysed the data with the ΔΔCt method. For each sample, we calculated normalization factors by averaging the (sample expression)/(average reference expression) ratios of three normalization genes, *act-1*, *tba-1*, and *ubc-2*. The reference sample was *EV(RNAi)*, wild-type, or 21% $O_2$, as appropriate. We used one-way or two-way ANOVA to calculate the statistical significance of gene expression changes and corrected for multiple comparisons using the Tukey method. Primers were tested on serial cDNA dilutions and analysed for PCR efficiency prior to use. All data originate from three or more independent biological repeats, and each PCR reaction was conducted in technical triplicate. Sequences of qRT-PCR primers are listed in the Key resources table.

## Analysis of fluorescent reporter lines via DIC and fluorescence microscopy

To analyse fluorescence in reporter lines, egg lays were performed on NGM plates seeded with OP50 or RNAi plates seeded with the appropriate HT115 RNAi culture. Worms were allowed to grow to adulthood. Plates were then kept in 21% $O_2$ or transferred to 0.5% $O_2$ for 4 hr and allowed to recover for 1 hr in normoxia before imaging to allow for GFP maturation; hence, these assays are technically post-hypoxia experiments. Worms were collected into M9 buffer containing 0.06% levamisole (Sigma L9756) for immobilization on 2% (w/v) agarose pads for microscopy. We captured images at ×10 magnification on a CoolSnap HQ camera (Photometrics) attached to a Zeiss Axioplan 2 compound microscope, followed by MetaMorph Imaging Software with Autoquant 3D digital deconvolution. For higher resolution images, we used the Hamamatsu ORCA-Flash4.0 LT+ Digital CMOS camera attached to a Leica SP8X confocal microscope at ×40 magnification. All images for the same experiment were captured at the same exposure time. Images were analysed using ImageJ software (https://imagej.nih.gov/ij/download.html), with fluorescence calculated by taking the difference of the background fluorescence from the mean intestinal or whole-worm fluorescence. For experiments imaging the *fmo-2p::gfp* and *acs-2p::gfp* reporters, intestinal fluorescence was measured. For experiments imaging *hpk-1p::gfp*, *nhr-49p::nhr-49::gfp*, *lgg-1p::gfp*, *atg-2p::gfp*, or *epg-6p::gfp*, whole-worm fluorescence was measured. For each experiment, at least three independent trials were performed with a minimum of 30 worms per condition.

## Autophagosome formation measurement

Autophagosome formation was measured by counting fluorescent foci in the hypodermal seam cells of animals expressing the translational LGG-1::GFP reporter (*Das et al., 2017*; *Zhang et al., 2015*). L3 worms were either kept in 21% $O_2$ or transferred to 0.5% $O_2$ for 5 hr. Worms were collected into 1 M $NaN_3$ for immobilization on 2% (w/v) agarose pads for microscopy, and the Leica SP8X microscope was used as above at ×63 magnification. For each experiment, at least three independent trials were performed with a minimum of 15 worms scored for GFP foci, totalling at least 110 individual seam cells per condition. For the micrographs shown in *Figure 4C*, image brightness and contrast were adjusted in ImageJ to best visualize the number of foci present in each seam cell; importantly, the

same adjustment was consistently applied throughout the whole image. The same brightness and contrast settings were applied to images within each genotype (i.e., comparing hypoxia-exposed and normoxia-exposed animals of the same genotype), but different settings were used for different genotypes.

## NHR-49 transgenic strains

To construct the *nhr-49p::nhr-49::gfp* containing plasmid, a 6.6 kb genomic fragment of the *nhr-49* gene (including a 4.4 kb coding region covering all *nhr-49* transcripts and a 2.2 kb promoter region) was cloned into the GFP expression vector pPD95.77 (Addgene #1495), as reported previously (*Ratnappan et al., 2014*). For generating tissue-specific constructs, the *nhr-49* promoter was replaced with tissue-specific promoters using SbfI and SalI restriction enzymes to create plasmids for expressing NHR-49 in the muscle (*myo-3p::nhr-49::gfp*), intestine (*gly-19p::nhr-49::gfp*), hypodermis (*col-12p::nhr-49::gfp*), and neurons (*rgef-1p::nhr-49::gfp*). 100 ng/µl of each plasmid was injected, along with pharyngeal muscle-specific *myo-2p::mCherry* as a co-injection marker (25 ng/µl), into the *nhr-49(nr2041)* mutant strain using standard methods (*Mello and Fire, 1995*). Strains were maintained by picking animals that were positive for both GFP and mCherry.

## Hypoxia sensitivity assays

Hypoxic conditions were maintained using continuous flow chambers, as previously described (*Fawcett et al., 2012*). Compressed gas tanks (5000 ppm $O_2$ balanced with $N_2$) were certified as standard to within 2% of indicated concentration from Praxair Canada (Delta, BC). Oxygen flow was regulated using Aalborg rotameters (Aalborg Instruments and Controls, Inc, Orangeburg, NY). Hypoxic chambers (and room air controls) were maintained in a 20°C incubator for the duration of the experiments.

For embryo survival assays, gravid first-day adult worms (picked as L4 the previous day) were allowed to lay eggs for 1–4 hr on plates seeded with 15 µl OP50 or appropriate HT115 RNAi bacteria the previous day. Adults were removed, and eggs were exposed to 0.5% $O_2$ for 24 hr or 48 hr. Animals were scored for developmental success (reached at least L4 stage) after being placed back into room air for 65 hr (following 24 hr exposure) or 42 hr (following 48 hr exposure). For RNAi survival assays, worms were grown for one generation from egg to adult on the appropriate HT115 RNAi bacteria before their progeny was used for the egg lay.

For larval development assays, gravid adult worms (picked as L4 the previous day) were allowed to lay eggs for 2 hr and kept at 20°C for 13–17 hr to allow hatching (egg lays for *nhr-49(nr2041)* strains with embryonic developmental delays were performed 2 hr earlier to ensure synchronization with wild-type worms). Freshly hatched L1 worms were transferred to plates seeded with 15 µl OP50 the previous day and exposed to 0.5% $O_2$ for 48 hr. Animals were placed back into room air and immediately scored for stage.

For all normoxia (21% $O_2$) comparison experiments, methods were as described above except plates were kept in room air for the duration (instead of being exposed to 0.5% $O_2$).

## Hydrogen sulfide sensitivity assay

Construction of $H_2S$ chambers was as previously described (*Fawcett et al., 2012*; *Miller and Roth, 2007*). In short, 5000 ppm $H_2S$ (balanced with $N_2$) was diluted with room air to a final concentration of 50 ppm and monitored with a custom $H_2S$ detector, as described (*Miller and Roth, 2007*). Compressed gas mixtures were obtained from Airgas (Seattle, WA) and certified as standard to within 2% of the indicated concentration. Survival assays were performed in three independent trials with 20 L4 animals picked onto OP50 seeded plates. Plates were exposed to 50 ppm $H_2S$ for 24 hr in a 20°C incubator, then returned to room air to score viability. Animals were scored 30 min after removal from $H_2S$, and plates with dead animals were re-examined after several hours to ensure animals had not reanimated.

## RNA sequencing

Synchronized L1 wild-type, *nhr-49(nr2041)*, and *hif-1(ia4)* worms were allowed to grow on OP50 plates to L4 stage, then either kept in 21% $O_2$ or transferred to 0.5% $O_2$ for 3 hr. RNA was isolated from whole worms as described above (immediately following hypoxia exposure). RNA integrity and quality were ascertained on a BioAnalzyer. Construction of strand-specific mRNA sequencing libraries and

sequencing (75 bp PET) on an Illumina HiSeq 2500 machine was done at the Sequencing Services facility of the Genome Sciences Centre, BC Cancer Agency, Vancouver BC, Canada (https://www.bcgsc.ca/services/sequencing-services). We sequenced >20 million reads per sample. The raw FASTQ reads obtained from the facility were trimmed using Trimmomatic version 0.36 (*Bolger et al., 2014*) with parameters LEADING:3 TRAILING:3 SLIDINGWINDOW:4:15 MINLEN:36. Next, the trimmed reads were aligned to the NCBI reference genome WBcel235 WS277 (https://www.ncbi.nlm.nih.gov/assembly/GCF_000002985.6/) using Salmon version 0.9.1 (*Patro et al., 2017*) with parameters -l A -p 8 --gcBias. Then, transcript-level read counts were imported into R and summed into gene-level read counts using tximport (*Soneson et al., 2015*). Genes not expressed at a level greater than one count per million (CPM) reads in at least three of the samples were excluded from further analysis. The gene-level read counts were normalized using the trimmed mean of M-values (TMM) in edgeR (*Robinson et al., 2010*) to adjust samples for differences in library size. Differential expression analysis was performed using the quasi-likelihood F-test with the generalized linear model (GLM) approach in edgeR (*Robinson et al., 2010*). Differentially expressed genes (DEGs) were defined as those with at least a two-fold difference between two individual groups at an FDR < 0.05. RNA-seq data have been deposited at NCBI Gene Expression Omnibus (https://www.ncbi.nlm.nih.gov/geo/) under the record GSE166788.

Functional enrichment analysis and visualization were performed using the Overrepresentation Analysis (ORA) module with the default parameters in easyGSEA in the eVITTA toolbox (https://tau.cmmt.ubc.ca/eVITTA/; input December 14, 2020; *Cheng et al., 2021*). easyVizR in the eVITTA toolbox was used to visualize the overlaps and disjoints in the DEGs (input December 14, 2020).

## Acknowledgements

We thank the Taubert, Miller, and Ghazi labs for comments on the manuscript. Some strains were provided by the CGC, which is funded by the NIH Office of Research Infrastructure Programs (P40 OD010440). Some strains were provided by The National BioResource Project (NBRP). Grant support was from The Canadian Institutes of Health Research (CIHR; PJT-153199 to ST), the Natural Sciences and Engineering Research Council of Canada (NSERC; RGPIN-2018-05133 to ST), the Cancer Research Society (CRS; to ST), and the National Institutes of Health (NIH; R01AG051659 and R56AG066682 to AG, R01AG044378 to DM). KRSD was supported by NSERC CGS-M, NSERC CGS-D, and BCCHR scholarships, and ST by a Canada Research Chair and a BCCHR IGAP award.

## Additional information

### Funding

| Funder | Grant reference number | Author |
|---|---|---|
| National Institutes of Health | R56AG066682 | Arjumand Ghazi |
| Natural Sciences and Engineering Research Council of Canada | RGPIN-2018-05133 | Stefan Taubert |
| National Institutes of Health | R01AG051659 | Arjumand Ghazi |
| Cancer Research Society | 22727 | Stefan Taubert |
| BC Children's Hospital Foundation | | Kelsie RS Doering<br>Stefan Taubert |
| Canada Research Chairs | | Stefan Taubert |
| National Institutes of Health | R01AG044378 | Dana L Miller |
| Natural Sciences and Engineering Research Council of Canada | | Kelsie RS Doering |

| Funder | Grant reference number | Author |
|---|---|---|
| Canadian Institutes of Health Research | PJT-153199 | Stefan Taubert |

The funders had no role in study design, data collection and interpretation, or the decision to submit the work for publication.

## Author contributions

Kelsie RS Doering, Conceptualization, Data curation, Formal analysis, Investigation, Methodology, Resources, Validation, Visualization, Writing – original draft, Writing – review and editing; Xuanjin Cheng, Data curation, Formal analysis, Investigation, Methodology, Visualization, Writing – original draft, Writing – review and editing; Luke Milburn, Formal analysis, Investigation, Methodology; Ramesh Ratnappan, Investigation, Methodology, Resources; Arjumand Ghazi, Conceptualization, Funding acquisition, Methodology, Project administration, Resources, Supervision, Writing – original draft, Writing – review and editing; Dana L Miller, Conceptualization, Funding acquisition, Investigation, Methodology, Project administration, Resources, Supervision, Writing – original draft, Writing – review and editing; Stefan Taubert, Conceptualization, Data curation, Funding acquisition, Project administration, Supervision, Writing – original draft, Writing – review and editing

## Author ORCIDs

Ramesh Ratnappan http://orcid.org/0000-0001-7055-9043
Dana L Miller http://orcid.org/0000-0003-3983-0493
Stefan Taubert http://orcid.org/0000-0002-2432-7257

## Decision letter and Author response

Decision letter https://doi.org/10.7554/eLife.67911.sa1
Author response https://doi.org/10.7554/eLife.67911.sa2

# Additional files

## Supplementary files

• Supplementary file 1. Summary of statistics of embryo hypoxia survival experiments. Statistical comparison of each genotype's ability to reach at least L4 following 24 hr exposure to 0.5% $O_2$ as embryo and then allowed to recover at 21% $O_2$ for 65 hr compared to worm embryos kept in 21% $O_2$ for 65 hr (two-way ANOVA corrected for multiple comparisons using the Tukey method).

• Supplementary file 2. Summary of statistics of larval hypoxia survival experiments. Statistical comparison of each genotype's ability to reach at least L4 stage from L1 stage following 48 hr exposure to 0.5% $O_2$ as embryos compared to animals kept in 21% $O_2$ for 48 hr.

• Supplementary file 3. Lists of genes regulated by hypoxia in various genotypes. (a) List of the 83 genes significantly upregulated more than twofold in 21% $O_2$ vs. 0.5% $O_2$ in wild-type and *hif-1(ia4)* animals, but not in *nhr-49(nr2041)* animals, i.e., *nhr-49*-dependent, *hif-1*-independent genes. (b) List of 139 genes significantly upregulated more than twofold in 21% $O_2$ vs. 0.5% $O_2$ in wild-type and *nhr-49(nr2041)* animals, but not in *hif-1(ia4)* animals, i.e., *hif-1*-dependent, *nhr-49*-independent genes. (c) List of 264 genes significantly upregulated more than twofold in 21% $O_2$ vs. 0.5% $O_2$ via RNA-seq in wild-type, *nhr-49(nr2041)*, and *hif-1(ia4)*.

• Transparent reporting form

• Source data 1. Source data for all indicated figures.

## Data availability

RNA-seq data have been deposited at NCBI Gene Expression Omnibus (https://www.ncbi.nlm.nih.gov/geo/) under the record GSE166788. All data generated or analysed during this study are included in the manuscript and Supplementary files. Raw data points from each N are shown in figures wherever possible. See transparent reporting form for details.

The following dataset was generated:

| Author(s) | Year | Dataset title | Dataset URL | Database and Identifier |
|---|---|---|---|---|
| Doering KRS, Cheng X, Taubert S | 2020 | NHR-49 controls a HIF-1 independent hypoxia adaptation pathway in *Caenorhabditis elegans* | https://www.ncbi.nlm.nih.gov/geo/query/acc.cgi?acc=GSE166788 | NCBI Gene Expression Omnibus, GSE166788 |

The following previously published dataset was used:

| Author(s) | Year | Dataset title | Dataset URL | Database and Identifier |
|---|---|---|---|---|
| Shen C, Nettleton D, Jiang M, Kim SK, Powell-Coffman JA | 2005 | Hypoxia response | https://www.ncbi.nlm.nih.gov/geo/query/acc.cgi?acc=GSE2836 | NCBI Gene Expression Omnibus, GSE2836 |

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
