## [Editor Report]

The highly conserved protein hypoxia-inducible factor (HIF) is a well-known regulator of animal responses to low-oxygen environments. Using the sophisticated genetic tools of the nematode *C. elegans*, this paper identifies a parallel mechanism, governed by a different conserved transcription factor, that also provides protection from hypoxia. These findings provide important new insight into the complex genetic architecture of the mechanisms that maintain organismal homeostasis in the face of environmental stress.

---

## [Decision Letter]

**Decision letter after peer review:**

Thank you for submitting your article "Nuclear Hormone Receptor NHR-49 controls a HIF-1-independent hypoxia adaptation pathway in *Caenorhabditis elegans*" for consideration by *eLife*. Your article has been reviewed by 3 peer reviewers, and the evaluation has been overseen by a Reviewing Editor and Piali Sengupta as the Senior Editor. The reviewers have opted to remain anonymous.

All three reviewers agree that your findings are interesting and have the potential to provide important new insight into hypoxia responses in *C. elegans*. However, there are a number of areas where the reviewers feel that the conclusions your paper draws are not fully supported by the data. Additionally, for the paper to provide the level of biological insight that would be appropriate for *eLife*, the reviewers feel some additional issues need to be addressed.

Based on the individual reviewer comments below, the following points must be addressed in a revised version.

(1) Autophagy. The reviewers feel that the link between hypoxia, nhr-49, and autophagy is one of the most important insights provided by your work. However, the role of autophagy in hypoxia, and its reliance on nhr-49, needs to be better substantiated.

a. As pointed out by Reviewers 2 and 3, the sample size for the experiments in Figure 3F and Supp Figure 4B are quite small. Together with the high variance in your measurements, it is unlikely that these experiments have the power to detect differences that could be present when comparing WT vs nhr-49 backgrounds. Please increase sample sizes here, ideally guided by power size calculations. As Reviewer 3 suggests, it may also be useful to consider altering the experimental conditions.

b. Please confirm the changes in expression of autophagy genes, and their dependence on nhr-49, by RT-PCR.

c. As suggested by reviewer 1, please examine autophagosome formation directly (e.g., with LGG-1::GFP) under hypoxic conditions, in WT, nhr-49, and hif-1 animals.

(2) Site of action of NHR-49. Reviewers 1 and 2 raise concerns about the interpretation and significance of these findings.

a. Please provide additional information on the site of NHR-49 induction by hypoxia, e.g., with high-magnification images.

b. Please confirm the specificity of the protective effects of NHR-49 overexpression by (i) examining body-wall-specific and pharyngeal-muscle-specific overexpression as suggested by Reviewer 2 OR (ii) using tissue-specific RNAi as suggested by Reviewer 1.

(3) RNAseq data. As suggested by Reviewer 1, please clarify the analysis presented in Figure 3A,B and, if possible, compare your results with other hypoxia transcriptome data that may have been previously reported.

(4) Interactions between NHR-67 and NHR-49.

a. As suggested by Reviewers 1 and 2, please examine transcriptional reporters for nhr-49 and nhr-67 in hypoxia and normoxia, in both wild-type and nhr-49/67 mutants.

b. Please investigate genetic interactions: is nhr-67 synthetic lethal with hif-1 and/or nhr-49 in the context of hypoxia survival?

c. As Reviewer 2 suggests, it would also be useful to determine whether specific subsets of hypoxia response genes are dependent on nhr-67, though this is not considered essential.

(5) Regulation of NHR-49 by HPK-1.

a. The reviewers consider the proposed mechanism to be quite speculative. If possible, it would be ideal for you to determine whether NHR-49 becomes phosphorylated under hypoxia, and whether this depends on hpk-1, as suggested by Reviewer 2, or whether the localization of NHR-49 changes under these conditions, as suggested by Reviewer 3. Alternatively, please modify the text and figures to make it clear that this part of your model has not been tested and that the regulation of nhr-49 by hpk-1 could be indirect.

b. To substantiate the connections between hpk-1, nhr-49, and autophagy, please determine whether the upregulation of autophagy genes requires hpk-1.

(6) In a number of places, the interpretation of your results should be softened. One example of this is lines 150-151 (Figure S1B). ~95% of WT animals are adults at 48 hr but 85% of nhr-49 mutants are still L3 and L4. However, you claim that "mutants did not significantly develop slower than wildtype". If so, a different test needs to be carried out, or sample sizes need to be increased, because it seems clear that there is a difference in developmental rate between these groups. Other examples where conclusions need to be tempered are provided in the reviewer comments below.

In addition to these points, please also consider the other issues raised by the reviewers (see below) as you prepare your revision.

*Reviewer #1 (Recommendations for the authors):*

1. To address Major Concern 2, the authors should confirm their RNA-seq results on new samples via qRT-PCR and also test whether autophagosomes are formed under hypoxic conditions (LGG-1::GFP), which should be dependent on nhr-49 but not hif-1 (if the authors model is true). The authors should expand their discussion to explain how increased autophagy would protect against hypoxia (metabolic shift for anaerobic respiration, decreased proteome reduces energetic demands, etc.).

2. To address Major Concerns 3 and 4, the authors should use tissue specific RNAi or hairpins to inactivate nhr-49 and assess whether loss in any one tissue compromises hypoxic response/survival. Additionally, if ectopic overexpression of NHR-49 within one tissue does indirectly protect other tissues through an adaptive response, one would predict that overexpressing NHR-49 in one tissue would mute the induction of NHR-49 a neighboring tissue after hypoxic treatment. This could be tested easily via combining their nhr-49p::NHR-49::GFP reporter and tissue specific overexpression of NHR-49. The authors should consider testing this possibility.

3. I strongly suggest avoiding the use of "worms" in formal manuscripts and use "animals" instead. It's jargon and diminishes the impact of *C. elegans* research when read by scientists using other systems.

*Reviewer #2 (Recommendations for the authors):*

– Figure 3F – These results are very crucial for the main conclusion of this study. Since RNAi produces more variable results across biological replicates, this experiment should be repeated with some of the autophagy mutants such as unc-51 etc.

– The sample sizes for the RNAi experiments in Figure 3F do not seem to be enough to capture the differences in hypoxia survival between WT and nhr-49 mutant conditions: 41% vs 25% (atg-7), 27% vs 13% (bec-1) and 38% vs 13% (epg-3). None of these differences were detected as statistically significant. This is likely due to n = 4 for RNAi treatments of the nhr-49 mutant and very high variability across biological replicates. The authors should at least double their sample sizes for these conditions and test whether there are any significant differences in hypoxia survival between WT and nhr-49 mutant when autophagy is inhibited. Similarly, in Supplementary Figure 4B, some of the autophagy gene RNAi conditions seem to reduce survival by 25% (with some data points showing greater than 50% reduction) even in normoxia. None of these differences were detected as statistically significant, which again is likely due to low sample sizes and high variability across replicates. These experiments need to be carefully designed so that there is enough statistical power to discern whether nhr-49 and autophagy are regulating hypoxia survival via the same or independent pathways and whether autophagy inhibition reduces survival also in normoxia.

– For tissue-specific rescue of nhr-49 (data shown in Figure 4B), the authors should also perform a body wall muscle-specific rescue experiment. Since nhr-49 is expressed in the muscles, it will be informative to know whether body wall muscle-specific nhr-49 function is sufficient to improve hypoxia survival at the organism level. In addition, a pharyngeal muscle-specific nhr-49 rescue might serve as a good negative control, where tissue-specific rescue might not be able to affect whole organism hypoxia survival.

– To confirm the bidirectional regulation between nhr-49 and nhr-67, the authors should create a gfp transcriptional reporter for nhr-67 and test whether nhr-49 inhibition affects nhr-67 expression in hypoxia and normoxia. The RNA-seq data shown in Figure 5A suggests that NHR-49 possibly regulates nhr-67 expression only in hypoxic conditions, but it will be informative to know whether this regulation is restricted to specific tissues. In comparison, NHR-67 seems to be regulating nhr-49 expression predominantly in the intestine in both normoxia and hypoxia (line 309).

– The physiological relevance of the bidirectional negative regulation between nhr-49 and nhr-67 is not clear. Inhibition of both nhr genes reduces survival in hypoxic conditions (Supplementary Figure 5D). While this study has found downstream targets of nhr-49 that promote hypoxia survival, it is not clear how nhr-67 fits into this regulation. In this regard, the authors should investigate two directions: (1) does nhr-67 also show synthetic lethal interactions with hif-1 and nhr-49 in the context of hypoxia survival (similar to Figure 2A)?, and (2) does nhr-67 inhibition in hypoxic conditions prevent the activation of a specific subset of hypoxia response genes, such as genes involved in detoxification and autophagy?

– The authors propose that survival during hypoxia is predominantly due to the upregulation of autophagy genes, but not due to upregulation of fmo-2 and acs-2 (lines 222-224). However, they have only tested whether hpk-1 removal alters the expression of fmo-2 and acs-2 genes (Figures 6A-E). To mechanistically elucidate whether HPK-1 and NHR-49 are acting via the same pathway to promote hypoxia survival (as suggested in Figure 6F), it should be tested whether hpk-1 mutation impairs the upregulation of autophagy genes during hypoxia.

– The role of HPK-1 in stabilizing NHR-49 during hypoxia is a novel finding that is of broad interest. In its current form, the proposed mechanism is quite speculative. Since this regulation is central to the proposed HIF-1-independent hypoxia response pathway, the authors should investigate whether NHR-49 is phosphorylated in low oxygen conditions in an HPK-1-dependent manner. A simple way to test this would be to use the nhr-49::gfp translational reporter strain and prepare whole worm extracts in normoxia and hypoxia conditions. An immunoblot using GFP antibody would show whether there is an upward shift in the NHR-49::GFP band size in hypoxic conditions, which should be reversible after CIP treatment of the whole worm extracts. This experiment can also be performed in the hpk-1 mutant background to validate where the NHR-49 phosphorylation is dependent on HPK-1. An alternative way to test this would be immunoprecipitating NHR-49::GFP protein in normoxia and hypoxic conditions and immunoblotting with phospho-Ser and phospho-Thr antibodies.

*Reviewer #3 (Recommendations for the authors):*

Question on results presented by author:

– Expression of fmo-2 increases in hypoxia (0.5%) in a nhr-49 and hif-1 dependent manner; approach was qRT-PCR and Pfmo-2::gfp reporter in wildtype and mutants.

– Figure 1A- large deviation in fmo-2 mRNA fold change in.5% hypoxia. Any experimental reasons why such is observed?

– The nhr-49 mutant does not survive 24 hrs 0.5% hypoxia (exposed as embryos) well or if exposed as L1 larvae, the post hypoxia larvae do not progress to L4 stage (24-hour exposure, allowed to develop to L4 stage). The nhr-1 mutant a similar survival rate as hif-1 mutants; whereas the double mutant was significantly lower. These data suggest separate pathways and that nhr-49 is required for adaptation to hypoxia.

– Were the nhr-49 L1 larvae exposed to hypoxia arrested or growing slow after hypoxia exposure (eventually reach L4/adult stage)?

– The hydrogen sulfide experiment separated out the role nhr-49 has in stress responses that hif-1 has.

– Authors took a transcriptomic approach (N2, nhr-49 and hif-1 mutants, normoxia/hypoxia). Results show 315 upregulated and 177 downregulated were dependent on nhr-49; 83 upregulated and 51 downregulated were hif-1 independent. Subset of gene expression responses to hypoxia are nhr-49 dependent. Based on transcriptomic data – of the 83 genes that require nhr-49 but not hif-1 for upregulation in hypoxia genes classified as autophagy and detoxification were enriched (Figure 3D).

– Supplementary files- present these as excel files or tables that communicate more data (wormbase ID, gene identification, gene name, gene description, classification based on GO terms or perhaps use wormCat and if category is significantly represented). This will be helpful to the community.

– Hypoxia induced acs-2 expression in a nhr-49 dependent manner is interesting, acs-2 expression was variable in the hif-1 mutant- ideas why? Due to variability in methodology (was RNA collected immediately after hypoxia exposure?)

– Important question- which nhr-49 regulated genes are critical for hypoxia survival. The fmo-2;acs-2 double mutant sensitive to hypoxia (embryo); individually hypoxia survival was overall not reduced. Reduction in autophagy genes via RNAi reduced hypoxia survival (atg-10, atg-7, atg-7, bec-1 and epg-3) in N2 and nhr-49 background.

– Line 235, line 238, Figure 3F- standard deviation not noted in text but the figure indicates it is large – authors have an indication as to why individual experiments vary, what is this due to? Have authors tried longer hypoxia exposures? This may intensify the hypoxia sensitivity and reduce variability in capacity to survive/develop after hypoxia exposure. I am not suggesting to redo all the hypoxia exposure experiments but something to keep in mind and consider to determine essential nature of the autophagy process in hypoxia.

– What is role of autopagy in adaptation to hypoxia? This section a bit disconnected from the fmo-2, acs-2 expression work.

– Line 227- did authors confirm expression changes, in nhr-49 dependent manner, of the autophagy genes using RT-PCR?

– nhr-49(et23) gain of function experiment is of interest.

– Elaborate and provide more details about what is known about this gf mutation. Unclear what a combined gain and loss of function properties indicate (environment dependent?).

– Line 280-281- unclear how other proteins that interact with NHR-49 were "studied"? What was done to identify NHR-67? May need to reword if no experiments conducted to identify NHR-67. Line 285, were other transcription factors that are upregulated in response to hypoxia screened through (RNAi) or did authors focus on nhr-67?

– nhr-67(RNAi) induced expression of fmo-2 and acs-2 (gfp reporters), indicating a negative regulation.

– Did nhr-67(RNAi) have an impact on the autophagy gene expression?

– Include text to provide better logic authors used to assess kinases as the next step in the study. The transition to these studies lacking. Why kinases? Provide text to transition and provide the rationale/relationship between oxidative stress and hypoxia and these particular kinases. Line 328, list the genes in a table with functional description or list in text.

– What is role of these kinases relative to NHR-49; epistasis suggests acting in same pathway. Figure 7- did authors observe any difference in the NHR-49 localization in the hpk-1 mutant? Impact on NHR-49 seemed minimal in hpk-1 mutant. Perhaps images to assess tissue expression better more convincing.

– Figure 8- data didn't demonstrate that HPK-1 dysfunction impacts autophagy genes; are reporters available or assess with RT-PCR.

– Line 545,546- may want to note in result that these assays are actually post-hypoxia experiments. The worm is recovering, unclear if the reporters are upregulated during hypoxia or if this is a post hypoxia expression.

– Wondering why the animal time in hypoxia differed between the RT-PCR/RNA-seq experiments and fluorescent reporter assays (3hrs, 4hrs, respectively; line 527 and 545).

– What was the genome coverage for RNA sequencing experiments? Can authors communicate if RNA isolation was immediately after hypoxia exposure or if animals were exposed to a normoxia recovery time.

---

## [Author Response]

Based on the individual reviewer comments below, the following points must be addressed in a revised version.(1) Autophagy. The reviewers feel that the link between hypoxia, nhr-49, and autophagy is one of the most important insights provided by your work. However, the role of autophagy in hypoxia, and its reliance on nhr-49, needs to be better substantiated.a. As pointed out by Reviewers 2 and 3, the sample size for the experiments in Figure 3F and Supp Figure 4B are quite small. Together with the high variance in your measurements, it is unlikely that these experiments have the power to detect differences that could be present when comparing WT vs nhr-49 backgrounds. Please increase sample sizes here, ideally guided by power size calculations. As Reviewer 3 suggests, it may also be useful to consider altering the experimental conditions.

We agree that the evidence in our initial manuscript did not convincingly link autophagy gene function to hypoxia survival and the *nhr-49* pathway. To overcome variability issues with RNAi, we instead studied functional requirements with mutants in two autophagy genes: *lgg-2* and *epg-6*. Analysis of *lgg-2* and *epg-6* single mutants and double mutants of these genes with the *nhr-49* mutant showed that mutation of each of these genes reduced hypoxia resistance, which was not exacerbated by concomitant *nhr-49* mutation. This provides strong support for our conclusion that autophagy contributes to hypoxia adaptation via the *nhr-49* pathway. These new results are presented in Figure 4E, F. The RNAi data have been moved to Figure 4—figure supplement 1F, G.

b. Please confirm the changes in expression of autophagy genes, and their dependence on nhr-49, by RT-PCR.

We appreciate the need to validate RNA-seq data and attempted to quantify gene expression changes with qRT-PCR. In this experiment, we observed a general increase in autophagy gene expression in wild-type worms exposed to hypoxia; however, these changes were not significant, likely because the increase in expression of each individual gene is relatively small (Author response image 1) .

**Author response image 1. sa2fig1:** qRT-PCR reveals that increase of autophagy gene expression is not significant. The graph indicates fold changes of mRNA levels in L4 stage wild-type animals exposed to room air (21% O_2_) or 0.5% O_2_ for 3 hr (n = 3). Statistics: two-way ANOVA corrected for multiple comparisons using the Tukey method.

As an alternative approach, we studied previously characterized transcriptional reporters (promoter-GFP fusions) of three autophagy genes and quantified their induction in normoxia vs. hypoxia following knockdown of *nhr-49*, *hif-1*, *hpk-1*, or *nhr-67*. The data show that all three autophagy genes are significantly induced approximately 1.5–2 fold in hypoxia, and that *nhr-49* and *hpk-1*, but not *nhr-67* and *hif-1*, are required for these inductions. This validates our RNA-seq data, and clearly links *nhr-49* and *hpk-1* to the regulation of autophagy genes in hypoxia. We have added these new data in Figure 4A, B and Figure 4—figure supplement 1B, C*.*

c. As suggested by reviewer 1, please examine autophagosome formation directly (e.g., with LGG-1::GFP) under hypoxic conditions, in WT, nhr-49, and hif-1 animals.

We thank the reviewer for this excellent suggestion. As requested, we used the LGG-1::GFP reporter to assess autophagosome formation in hypoxia in the WT, *nhr-49*, and *hif-1* mutant backgrounds. Consistent with our gene regulation and functional data, we observed a clear requirement for *nhr-49*, but not *hif-1*, in the upregulation of autophagosome formation in hypoxia (new Figure 4C, D).

In addition, we performed a similar experiment using RNAi for *hpk-1* and *nhr-*67. This analysis revealed that *hpk-1*, but not *nhr-67*, is required for LGG-1::GFP induction in hypoxia, (Figure 4G).

Collectively we believe that these experiments provide strong support of one of the most important conclusions of our study: that *nhr-49* and *hpk-1*, but not *hif-1*, are required for autophagy induction and hence survival in hypoxia.

(2) Site of action of NHR-49. Reviewers 1 and 2 raise concerns about the interpretation and significance of these findings.a. Please provide additional information on the site of NHR-49 induction by hypoxia, e.g., with high-magnification images.

As requested, we have included additional high-magnification images assessing NHR-49 induction in hypoxia. This shows that NHR-49::GFP is induced in the head region, in the hypodermis, and in the intestine (new Figure 2D and Figure 2—figure supplement 1E, F).

b. Please confirm the specificity of the protective effects of NHR-49 overexpression by (i) examining body-wall-specific and pharyngeal-muscle-specific overexpression as suggested by Reviewer 2 OR (ii) using tissue-specific RNAi as suggested by Reviewer 1.

i. We were able to confirm a similar protective effect of NHR-49 rescue in body-wall muscle as in other tissues; these data have been added to Figure 5B. However, we were not able to obtain a pharynx-specific rescue strain as attempts to generate this strain failed for unknown reasons.

ii. We appreciate the suggestion of the reviewer for this interesting experiment and tried it as requested. However, in our hypoxia/normoxia survival assays, the tissue-specific RNAi strains featured baseline sickness and large variability, which prevented us from reliably detecting effects of *nhr-49* RNAi on hypoxia survival (see Author response image 2). For this reason, we did not include these results in the manuscript.

**Author response image 2. sa2fig2:** Tissue specific RNAi experiments resulted in highly variable hypoxia resistance. The graph shows average population survival of *rde-1(ne219)* (control, RNAi deficient), OLB11 *(rde-1(ne219);(pOLB11(elt-2p::rde-1) + pRF4(rol-6(su1006))*); intestine-specific RNAi; McGhee et al., 2009), NR350 (*rde-1(ne219) V; kzIs20 [hlh-1p::rde-1 + sur-5p::NLS::GFP]*; muscle-specific RNAi; Qadota et al., 2007), and NR222 (*rde-1(ne219) V; kzIs9 [(pKK1260) lin-26p::NLS::GFP + (pKK1253) lin-26p::rde-1 + rol-6(su1006)]*; hypodermis-specific RNAi; Qadota et al., 2007) strains grown on control empty vector RNAi (*EV(RNAi)*) or *nhr-49(RNAi)*. Worm embryos were exposed for 24 hr to 0.5% O_2_ and then allowed to recover at 21% O_2_ for 65 hr, and then scored for ability to reach at least the L3 stage (four repeats totalling >100 individual animals per genotype) (left). As control, we determined developmental success of the same strains on the same RNAi conditions grown in normoxia (right). Statistics: ordinary one-way ANOVA corrected for multiple comparisons using the Tukey method.

(3) RNAseq data. As suggested by Reviewer 1, please clarify the analysis presented in Figure 3A,B and, if possible, compare your results with other hypoxia transcriptome data that may have been previously reported.

We apologize for the confusion in our analysis and description. We have revised the figure legends of Figure 3A, B and the corresponding text on lines 290-295 to better explain which genes belong in which groups.

We also appreciate the idea of comparing our RNA-seq data to previously generated *C. elegans* hypoxia transcriptome data. Surprisingly, we were not able to find many such datasets. RNA-seq data of wild-type worms exposed to hypoxia have been published (Vozdek et al., 2018)*.* However, the data were not deposited in GEO or similar databases and we were unable to compare the Vozdek dataset with ours since p-values were not available in the former. The only other paper that we found with hypoxia-normoxia transcriptome data is Shen et al., 2005. Comparing our gene list to the list from this paper, we found that 24 of 110 genes identified by Shen are also hypoxia-inducible in our study (new Figure 3—figure supplement 1B). This may appear low, but likely reflects the fact that Shen et al., studied different conditions and used different analysis and processing approaches (L3 larval stage by Shen, L4 by us; 0.1% O_2_ by Shen, 0.5% O_2_ by us; microarrays by Shen, RNA-seq by us). We note that our RNA-seq successfully identified several genes that have been shown by other studies to be hypoxia inducible in *C. elegans*, including *nhr-57*, *egl-9*, *fmo-2*, and F22B5.4 (Bishop et al., 2004; Shen et al., 2005). In sum, we are confident that our hypoxia RNA-seq analysis is of high quality and hope that it will serve as a reference study for future investigations in this area.

(4) Interactions between NHR-67 and NHR-49.a. As suggested by Reviewers 1 and 2, please examine transcriptional reporters for nhr-49 and nhr-67 in hypoxia and normoxia, in both wild-type and nhr-49/67 mutants.

We thank the reviewers for this suggestion. To examine *nhr-67* expression following knockdown of *nhr-49*, we used the MU1268 strain, which contains an extrachromosomal array wherein the promoter and the first three exons of *nhr-67* are linked to GFP (Gissendanner et al., 2004). However, although we did detect significant increase of *nhr-67p::gfp* in hypoxia in both *EV(RNAi)* and *nhr-49(RNAi)* conditions, GFP expression in these worms was extremely low, even after *nhr-49* knockdown in hypoxia, and thus we would prefer not to include these data in the manuscript (Author response image 3) .

**Author response image 3. sa2fig3:** Analysis of *nhr-67* and *nhr-49* expression. (Left) The graph shows the quantification of GFP levels in *nhr67p::gfp* animals in normoxia or following 4 hr exposure to 0.5% O_2_ and 1 hr recovery in 21% O_2_, on control empty vector RNAi (*EV(RNAi)*), *nhr-49(RNAi)*, or *hif-1(RNAi)* (three repeats totalling >30 individual animals per genotype). ** p <0.01 (two-way ANOVA corrected for multiple comparisons using the Tukey method). (Right) The graph indicates relative levels of *nhr-49* mRNA in L4 control empty vector RNAi (*EV(RNAi)*) and *nhr-49(RNAi)* animals exposed to room air (21% O_2_) or 0.5% O_2_ for 3 hr (n = 3). Statistics: two-way ANOVA corrected for multiple comparisons using the Tukey method.

To assess *nhr-49* expression following knockdown of *nhr-67*, we attempted to generate an *nhr-49p::gfp* transcriptional reporter. However, numerous attempts by two labs (Taubert, Ghazi) failed to generate the appropriate expression vector. As an alternative approach, we assessed *nhr-49* mRNA levels in hypoxia and normoxia in the control (empty vector, EV) and *nhr-67(RNAi)* backgrounds. Consistent with our conclusion that NHR-49 is regulated post-transcriptionally in response to stress, mRNA expression level was similar in all these conditions (Author response image 3).

b. Please investigate genetic interactions: is nhr-67 synthetic lethal with hif-1 and/or nhr-49 in the context of hypoxia survival?

As requested, we tested genetic interactions between these three transcriptional regulators. Specifically, we quantified embryo survival in hypoxia in *nhr-49(-);nhr67(RNAi)* and *hif-1(-);nhr67(RNAi)* worm embryos. In line with our previous data, these new experiments showed that *nhr-67* acts within the *nhr-49* response pathway but shows synthetic lethality with *hif-1* (see updated Figure 6—figure supplement 1D).

c. As Reviewer 2 suggests, it would also be useful to determine whether specific subsets of hypoxia response genes are dependent on nhr-67, though this is not considered essential.

We appreciate the interest in the role *nhr-67* plays in the hypoxia response. In our revised manuscript, we include new data showing that *nhr-67* does not appear to play a regulatory role in autophagy during hypoxia, via both transcriptional reporter assays and autophagosome formation (LGG-1::GFP foci) (new Figures 4A, B, G, Figure 4—figure supplement 1B, C). We have also shown that *nhr-67* is important in the negative regulation of *fmo-2* and *acs-2* (Figures 6B-E). Further exploration of genes and processes regulated by *nhr-67* via RNA-seq analysis will be an interesting future direction.

In sum, although some of these experiments were not successful, the available data agree with our conclusion – that NHR-67 acts in the new NHR-49 pathway and in parallel to HIF-1.

(5) Regulation of NHR-49 by HPK-1.a. The reviewers consider the proposed mechanism to be quite speculative. If possible, it would be ideal for you to determine whether NHR-49 becomes phosphorylated under hypoxia, and whether this depends on hpk-1, as suggested by Reviewer 2, or whether the localization of NHR-49 changes under these conditions, as suggested by Reviewer 3. Alternatively, please modify the text and figures to make it clear that this part of your model has not been tested and that the regulation of nhr-49 by hpk-1 could be indirect.

We attempted to investigate how HPK-1 may regulate NHR-49, but pilot experiments to study effects on NHR-49 did not reliably identify bands that might reflect phosphorylated NHR-49. We have instead substantially toned down our conclusions to indicate the speculative nature of this part of our model (see text line 747-750 and Figure 9).

b. To substantiate the connections between hpk-1, nhr-49, and autophagy, please determine whether the upregulation of autophagy genes requires hpk-1.

We agree that the role of *hpk-1* in autophagy regulation needed further testing, and have done as requested. As noted in the response to comment 1b, hypoxia induction of autophagy gene expression (promoter-GFP reporters) and of autophagosome formation (LGG-1::GFP foci) showed strong reliance on *hpk-1*, supporting our core findings (Figure 4A, B, G, Figure 4—figure supplement 1B, C).

(6) In a number of places, the interpretation of your results should be softened. One example of this is lines 150-151 (Figure S1B). ~95% of WT animals are adults at 48 hr but 85% of nhr-49 mutants are still L3 and L4. However, you claim that "mutants did not significantly develop slower than wildtype". If so, a different test needs to be carried out, or sample sizes need to be increased, because it seems clear that there is a difference in developmental rate between these groups. Other examples where conclusions need to be tempered are provided in the reviewer comments below.

We apologize for overinterpretation and have changed the wording in this (line 187-209) and other instances.

In addition to these points, please also consider the other issues raised by the reviewers (see below) as you prepare your revision.Reviewer #1 (Recommendations for the authors):1. To address Major Concern 2, the authors should confirm their RNA-seq results on new samples via qRT-PCR and also test whether autophagosomes are formed under hypoxic conditions (LGG-1::GFP), which should be dependent on nhr-49 but not hif-1 (if the authors model is true). The authors should expand their discussion to explain how increased autophagy would protect against hypoxia (metabolic shift for anaerobic respiration, decreased proteome reduces energetic demands, etc.).

As requested, we have performed validation experiments for our RNA-seq and studied autophagosomes, as detailed above (Editor Comment #1). To address putative protective effects of autophagy in hypoxia, we have edited the text in the discussion on lines 892-900.

2. To address Major Concerns 3 and 4, the authors should use tissue specific RNAi or hairpins to inactivate nhr-49 and assess whether loss in any one tissue compromises hypoxic response/survival. Additionally, if ectopic overexpression of NHR-49 within one tissue does indirectly protect other tissues through an adaptive response, one would predict that overexpressing NHR-49 in one tissue would mute the induction of NHR-49 a neighboring tissue after hypoxic treatment. This could be tested easily via combining their nhr-49p::NHR-49::GFP reporter and tissue specific overexpression of NHR-49. The authors should consider testing this possibility.

With regards to the use of tissue specific RNAi, please see above (Editor Comment # 2b).

With regards to the second proposed experiment, this is not feasible because the tissue-specific NHR-49 overexpression strains also all bear the NHR-49 GFP fusion as well as the same transgenic marker; these existing strains can therefore not be used to generate the proposed double transgenic strains by crossing. In addition, the nhr-49p::NHR-49::GFP strain already substantially overexpresses NHR-49, and we hesitate introducing another gain-of-function transgene into this genetic context; interpretation of resulting data might be tricky, as pointed out by the reviewer themselves. We have therefore opted not to pursue this line of investigation.

3. I strongly suggest avoiding the use of "worms" in formal manuscripts and use "animals" instead. It's jargon and diminishes the impact of *C. elegans* research when read by scientists using other systems.

We appreciate the comment and have corrected the terminology throughout the manuscript.

Reviewer #2 (Recommendations for the authors):– Figure 3F – These results are very crucial for the main conclusion of this study. Since RNAi produces more variable results across biological replicates, this experiment should be repeated with some of the autophagy mutants such as unc-51 etc.

We thank this reviewer for this suggestion. As requested, we have done these experiments using mutants in autophagy genes. Please see above (Editor Comment #1a) for details.

– The sample sizes for the RNAi experiments in Figure 3F do not seem to be enough to capture the differences in hypoxia survival between WT and nhr-49 mutant conditions: 41% vs 25% (atg-7), 27% vs 13% (bec-1) and 38% vs 13% (epg-3). None of these differences were detected as statistically significant. This is likely due to n = 4 for RNAi treatments of the nhr-49 mutant and very high variability across biological replicates. The authors should at least double their sample sizes for these conditions and test whether there are any significant differences in hypoxia survival between WT and nhr-49 mutant when autophagy is inhibited. Similarly, in Supplementary Figure 4B, some of the autophagy gene RNAi conditions seem to reduce survival by 25% (with some data points showing greater than 50% reduction) even in normoxia. None of these differences were detected as statistically significant, which again is likely due to low sample sizes and high variability across replicates. These experiments need to be carefully designed so that there is enough statistical power to discern whether nhr-49 and autophagy are regulating hypoxia survival via the same or independent pathways and whether autophagy inhibition reduces survival also in normoxia.

We agree and in our revised manuscript have redone this analysis with autophagy mutant strains, which strongly supports our conclusions. Please see above for details (Editor Comment #1a).

– For tissue-specific rescue of nhr-49 (data shown in Figure 4B), the authors should also perform a body wall muscle-specific rescue experiment. Since nhr-49 is expressed in the muscles, it will be informative to know whether body wall muscle-specific nhr-49 function is sufficient to improve hypoxia survival at the organism level. In addition, a pharyngeal muscle-specific nhr-49 rescue might serve as a good negative control, where tissue-specific rescue might not be able to affect whole organism hypoxia survival.

As requested we have added experiments with a body wall muscle-specific rescue strain, which also provides functional rescue. Please see above for details (Editor Comment # 2b).

– To confirm the bidirectional regulation between nhr-49 and nhr-67, the authors should create a gfp transcriptional reporter for nhr-67 and test whether nhr-49 inhibition affects nhr-67 expression in hypoxia and normoxia. The RNA-seq data shown in Figure 5A suggests that NHR-49 possibly regulates nhr-67 expression only in hypoxic conditions, but it will be informative to know whether this regulation is restricted to specific tissues. In comparison, NHR-67 seems to be regulating nhr-49 expression predominantly in the intestine in both normoxia and hypoxia (line 309).

As requested we have further investigated the relationship between NHR-49 and NHR-67. Please see above (Editor Comment #4a).

– The physiological relevance of the bidirectional negative regulation between nhr-49 and nhr-67 is not clear. Inhibition of both nhr genes reduces survival in hypoxic conditions (Supplementary Figure 5D). While this study has found downstream targets of nhr-49 that promote hypoxia survival, it is not clear how nhr-67 fits into this regulation. In this regard, the authors should investigate two directions: (1) does nhr-67 also show synthetic lethal interactions with hif-1 and nhr-49 in the context of hypoxia survival (similar to Figure 2A)?, and (2) does nhr-67 inhibition in hypoxic conditions prevent the activation of a specific subset of hypoxia response genes, such as genes involved in detoxification and autophagy?

We further examined crosstalk between *nhr-67* and the other two regulators; in response to (1), our data suggest that *nhr-67* loss interacts genetically with *hif-1* loss, showing that they act in separate pathways; in response to (2), we observed that depletion of *nhr-67* did not affect the induction of autophagy gene reporters or autophagosomes by hypoxia. For details, please see above (Editor Comment #4b, 4c).

– The authors propose that survival during hypoxia is predominantly due to the upregulation of autophagy genes, but not due to upregulation of fmo-2 and acs-2 (lines 222-224). However, they have only tested whether hpk-1 removal alters the expression of fmo-2 and acs-2 genes (Figures 6A-E). To mechanistically elucidate whether HPK-1 and NHR-49 are acting via the same pathway to promote hypoxia survival (as suggested in Figure 6F), it should be tested whether hpk-1 mutation impairs the upregulation of autophagy genes during hypoxia.

As requested, we not only tested whether *hpk-1* regulates the expression of autophagy related genes in hypoxia, but also assessed whether it is required for autophagosome formation, both of which excitingly is the case, in agreement with our model. For details, please see above (Editor Comment #1b, 3b).

– The role of HPK-1 in stabilizing NHR-49 during hypoxia is a novel finding that is of broad interest. In its current form, the proposed mechanism is quite speculative. Since this regulation is central to the proposed HIF-1-independent hypoxia response pathway, the authors should investigate whether NHR-49 is phosphorylated in low oxygen conditions in an HPK-1-dependent manner. A simple way to test this would be to use the nhr-49::gfp translational reporter strain and prepare whole worm extracts in normoxia and hypoxia conditions. An immunoblot using GFP antibody would show whether there is an upward shift in the NHR-49::GFP band size in hypoxic conditions, which should be reversible after CIP treatment of the whole worm extracts. This experiment can also be performed in the hpk-1 mutant background to validate where the NHR-49 phosphorylation is dependent on HPK-1. An alternative way to test this would be immunoprecipitating NHR-49::GFP protein in normoxia and hypoxic conditions and immunoblotting with phospho-Ser and phospho-Thr antibodies.

We attempted to investigate NHR-49 phosphorylation but were unable to conclusively identify phosphorylated NHR-49. In line with the suggestion of the editor, we have therefore tempered our interpretation of the role of HPK-1 in NHR-49 regulation. For details, please see above (Editor Comment #5a).

Reviewer #3 (Recommendations for the authors):Question on results presented by author:– Expression of fmo-2 increases in hypoxia (0.5%) in a nhr-49 and hif-1 dependent manner; approach was qRT-PCR and Pfmo-2::gfp reporter in wildtype and mutants.– Figure 1A- large deviation in fmo-2 mRNA fold change in.5% hypoxia. Any experimental reasons why such is observed?

We agree with the reviewer that these deviations are rather large; however, importantly, they are statistically significant. We do not have a clear explanation for the large variability – worms are all at the same stage, and are exposed for same time. Potentially, post-hypoxia harvesting procedures vary by minutes such that changes manifest due to normoxia exposure during the washing and harvesting procedure.

– The nhr-49 mutant does not survive 24 hrs 0.5% hypoxia (exposed as embryos) well or if exposed as L1 larvae, the post hypoxia larvae do not progress to L4 stage (24-hour exposure, allowed to develop to L4 stage). The nhr-1 mutant a similar survival rate as hif-1 mutants; whereas the double mutant was significantly lower. These data suggest separate pathways and that nhr-49 is required for adaptation to hypoxia.– Were the nhr-49 L1 larvae exposed to hypoxia arrested or growing slow after hypoxia exposure (eventually reach L4/adult stage)?

We did not test whether arrested worms eventually reach adulthood. Anecdotally, we have observed that worms able reach the L3 or L4 stage eventually develop into adults, whereas animals arrested at the L1 or L2 stages remain permanently arrested.

– The hydrogen sulfide experiment separated out the role nhr-49 has in stress responses that hif-1 has.– Authors took a transcriptomic approach (N2, nhr-49 and hif-1 mutants, normoxia/hypoxia). Results show 315 upregulated and 177 downregulated were dependent on nhr-49; 83 upregulated and 51 downregulated were hif-1 independent. Subset of gene expression responses to hypoxia are nhr-49 dependent. Based on transcriptomic data – of the 83 genes that require nhr-49 but not hif-1 for upregulation in hypoxia genes classified as autophagy and detoxification were enriched (Figure 3D).– Supplementary files- present these as excel files or tables that communicate more data (wormbase ID, gene identification, gene name, gene description, classification based on GO terms or perhaps use wormCat and if category is significantly represented). This will be helpful to the community.

We agree with the reviewer that these files will be important for the community and provide them in our revised manuscript as Supplementary File 3.

– Hypoxia induced acs-2 expression in a nhr-49 dependent manner is interesting,acs-2 expression was variable in the hif-1 mutant- ideas why? Due to variability in methodology (was RNA collected immediately after hypoxia exposure?)

We do not know why *acs-2* expression is somewhat variable. Methodology was kept as consistent between individual experimental repeats as possible.

– Important question- which nhr-49 regulated genes are critical for hypoxia survival. The fmo-2;acs-2 double mutant sensitive to hypoxia (embryo); individually hypoxia survival was overall not reduced. Reduction in autophagy genes via RNAi reduced hypoxia survival (atg-10, atg-7, atg-7, bec-1 and epg-3) in N2 and nhr-49 background.– Line 235, line 238, Figure 3F- standard deviation not noted in text but the figure indicates it is large – authors have an indication as to why individual experiments vary, what is this due to? Have authors tried longer hypoxia exposures? This may intensify the hypoxia sensitivity and reduce variability in capacity to survive/develop after hypoxia exposure. I am not suggesting to redo all the hypoxia exposure experiments but something to keep in mind and consider to determine essential nature of the autophagy process in hypoxia.

We agree that it was important to provide further evidence showing that autophagy genes are important for hypoxia adaptation and have done as requested. For details, please see above (Editor Comment #1a).

We appreciate the idea to performing longer hypoxia survival assays to reduce variability, and we have done so in Figure 5A. However, this also reduces WT survival, and therefore is not suitable for all assays.

– What is role of autopagy in adaptation to hypoxia? This section a bit disconnected from the fmo-2, acs-2 expression work.

We agree that this needed to be evaluated further and have extensively done so. Please see above (Reviewer #1, Major Suggestion #1).

– Line 227- did authors confirm expression changes, in nhr-49 dependent manner, of the autophagy genes using RT-PCR?

As requested, in our revision, we confirm gene expression changes by means of promoter-GFP reporter analysis. For details, please see above (Editor Comment #1b).

– nhr-49(et23) gain of function experiment is of interest.– Elaborate and provide more details about what is known about this gf mutation. Unclear what a combined gain and loss of function properties indicate (environment dependent?).

We agree that this was somewhat unclear. This mutant has been characterized in detail in Lee et al., 2016. We have added a reference to that extent.

– Line 280-281- unclear how other proteins that interact with NHR-49 were "studied"? What was done to identify NHR-67?

We apologize for the oversight. Interaction between NHR-49 and NHR-67 was identified in a previous study (Reece-Hoyes et al., 2013). We have added text to that extent on lines 523-526.

May need to reword if no experiments conducted to identify NHR-67. Line 285, were other transcription factors that are upregulated in response to hypoxia screened through (RNAi) or did authors focus on nhr-67?

We apologize that this was not clear. We examined in our RNA-seq data the expression of TFs known to interact with NHR-49 (from Reece-Hoyes et al., 2013) and noticed that *nhr-67* showed an interesting pattern of regulation. We have clarified this further in the manuscript on lines 523-526.

– nhr-67(RNAi) induced expression of fmo-2 and acs-2 (gfp reporters), indicating a negative regulation.– Did nhr-67(RNAi) have an impact on the autophagy gene expression?

As requested we addressed whether *nhr-67* loss affects autophagy gene regulation (new Figure 4A, B, Figure 4—figure supplement 1B, C). For details, please see above (Editor Comment #1b).

– Include text to provide better logic authors used to assess kinases as the next step in the study. The transition to these studies lacking. Why kinases? Provide text to transition and provide the rationale/relationship between oxidative stress and hypoxia and these particular kinases. Line 328, list the genes in a table with functional description or list in text.

We apologize for the unclear rationale. We investigated kinases because they are known to act upstream of many stress responsive transcription factors. We have added text to this extent on lines 633-636.

With regards to the genes tested, we did not screen them for hypoxia related action. Instead, we performed a separate screen for a different project on NHR-49 in stress response pathways (manuscript currently in preparation). For this reason, we prefer not to disclose the full list of genes identified in the present manuscript, wherein we focused solely on HPK-1.

– What is role of these kinases relative to NHR-49; epistasis suggests acting in same pathway. Figure 7- did authors observe any difference in the NHR-49 localization in the hpk-1 mutant? Impact on NHR-49 seemed minimal in hpk-1 mutant. Perhaps images to assess tissue expression better more convincing.

We agree and provide higher magnification images. For details, please see above (Editor Comment #2a).

– Figure 8- data didn't demonstrate that HPK-1 dysfunction impacts autophagy genes; are reporters available or assess with RT-PCR.

As requested, we used transcriptional reporters to provide comprehensive evidence that HPK-1 affects the regulation of autophagy genes in hypoxia as well as autophagosome formation. For detail, please see above (Editor Comment #1a, b).

– Line 545,546- may want to note in result that these assays are actually post-hypoxia experiments. The worm is recovering, unclear if the reporters are upregulated during hypoxia or if this is a post hypoxia expression.

We appreciate the astute comment of the reviewer. In pilot experiments, anecdotally, we observe slight upregulation directly after hypoxia; however, most of the GFP is made only once the worms return to normoxia because many of the translation steps involved in GFP production require oxygen. In addition, oxygen is required for maturation of the GFP fluorophore. We have added text to this extent in the methods on lines 1014-1016.

– Wondering why the animal time in hypoxia differed between the RT-PCR/RNA-seq experiments and fluorescent reporter assays (3hrs, 4hrs, respectively; line 527 and 545).

We used a longer timepoint in the experiments with GFP-reporters because not only mRNA but also GFP protein has to be produced. We performed several pilot experiments and settled on the incubation times where we observed the most difference for each type of experiment.

– What was the genome coverage for RNA sequencing experiments? Can authors communicate if RNA isolation was immediately after hypoxia exposure or if animals were exposed to a normoxia recovery time.

Worms were frozen immediately after hypoxia, with no recovery time. We sequenced >20 million reads per sample. Details of this procedure have been added to the methods section on lines 1111-1112.